# Not All Thoughts are Generated Equal: Efficient LLM Reasoning via Synergizing-Oriented Multi-Turn Reinforcement Learning

## Abstract

Compressing long chain-of-thought (CoT) from large language models (LLMs) is an emerging strategy to improve the reasoning efficiency of LLMs. Despite its promising benefits, existing studies equally compress all thoughts within a long CoT, hindering more concise and effective reasoning. To this end, we first investigate the importance of different thoughts by examining their effectiveness and efficiency in contributing to reasoning through automatic long CoT chunking and Monte Carlo rollouts. Building upon the insights, we propose a theoretically bounded metric to jointly measure the effectiveness and efficiency of different thoughts. We then propose **Long⊗Short**, an efficient reasoning framework that enables two LLMs to collaboratively solve the problem: a long-thought LLM for more effectively generating important thoughts, while a short-thought LLM for efficiently generating remaining thoughts. Specifically, we begin by synthesizing a small amount of cold-start data to fine-tune LLMs for long-thought and short-thought reasoning styles, respectively. Furthermore, we propose a synergizing-oriented multi-turn reinforcement learning, focusing on the model self-evolution and collaboration between long-thought and short-thought LLMs. Experimental results show that our method enables Qwen2.5-7B and Llama3.1-8B to **achieve comparable performance** compared to DeepSeek-R1-Distill-Qwen-7B and DeepSeek-R1-Distill-Llama-8B, while **reducing token length by over 80%** across the MATH500, AIME24/25, AMC23, and GPQA Diamond benchmarks.

## 1 Introduction

Long Chain-of-Thought (CoT) reasoning aims to achieve test-time scaling Snell et al. (2024) by increasing the length of the CoT Wei et al. (2022) reasoning process. Recently, with the advancement of LLMs, such as OpenAI o-1 and DeepSeek-R1 Guo et al. (2025), long CoT reasoning has demonstrated significant improvements on various challenging tasks, such as mathematics Olympiad Hendrycks et al. (2021a), professional coding Jain et al. (2024) and scientific reasoning Rein et al. (2024) and so on. Long CoT reasoning has gradually become a key capacity of LLMs. Despite these advances, the excessive token length Jin et al. (2024) of long CoT reasoning remains a major bottleneck, limiting its effectiveness and practical applicability.

In recent literature, considerable efforts have been made to compress reasoning token length while maintaining performance Cheng et al. (2024). Overall, existing solutions can be divided into three categories: *training-free*, *supervised fine-tuning (SFT) based*, and *reinforcement learning (RL) based*. Training-free methods prompt LLMs to use limit word count to answer given question Kumar et al. (2025), or explicitly set the maximum token-budget Han et al. (2024) in inference stage, e.g., Qwen3Team (2025). SFT-based approaches often rely on rejection sampling Yuan et al. (2023), which selects correct but shortest reasoning trajectories through repetitive sampling. For example, UPFT Ji et al. (2025) and C3oT Kang et al. (2025) directly use selected trajectories to fine-tune LLMs for reasoning token length compression. RL-based methods employ offline or online learning strategy for CoT-length preference learning. On the one hand, offline learning methods like DAST Shen et al. (2025a) and O1-Pruner Luo et al. (2025) construct offline pairwise samples by treating the correct shorter response as positive and the correct longer response as negative, and applies RL algorithms Rafailov et al. (2023) to optimize length preferences. On the other hand, online RL methods such as

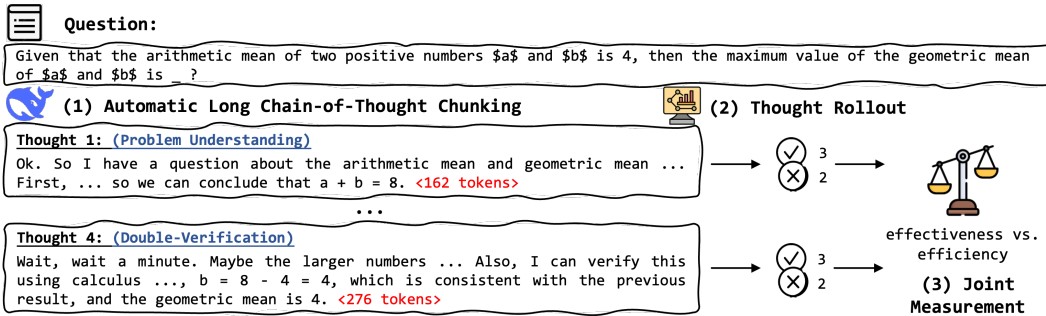

Figure 1: The workflow of how to investigate the importance of thoughts within a long CoT.

Kimi k1.5 Team et al. (2025) and DAPO Yu et al. (2025a) incorporate length-aware reward functions to penalize unnecessary reasoning steps. However, these efforts **equally compress all thoughts within a long CoT** without considering differences in their importance, e.g., as shown in Figure 1, "double-verification" thought might be less important than the "problem understanding" as it doesn't bring extra accuracy but costs more tokens, which hinders more concise and effective reasoning.

To this end, we first investigate the importance of different thoughts within a long CoT by examining their contributions to reasoning. Specifically, we use a three-step approach shown in Figure 1: *(1) Automatic Long CoT Chunking:* Unlike existing work that relies thought templates Yang et al. (2025a); Wan et al. (2025); Aytes et al. (2025) or split characters (e.g., '\n\n' Zhong et al. (2025) or 'wait' Lu et al. (2025)), we propose to automatically split long CoT distilled from recent advanced LLMs (e.g., DeepSeek-R1) into multiple thought chunks, facilitating thought-level analysis. *(2) Thought Rollout:* Different from previous works Xia et al. (2025) that use token probability to quantify, we run Monte Carlo rollouts Snell et al. (2024) of each thought to approximate their effectiveness and efficiency contribution for the reasoning process. We find that front thoughts tend to yield a higher contribution, even enabling Qwen2.5-7B to achieve a performance comparable to its distillation version, DeepSeek-R1-Distill-Qwen-7B, under a zero-shot prompting strategy. *(3) Joint Measurement:* Building upon the insights, we propose a joint metric that simultaneously measures both the effectiveness and efficiency contribution of each thought. Our theoretical analysis proves that the deviation of the proposed metric from the optimal one is upper-bounded.

Based on the above thought measurement approach, we propose **Long⊗Short**, an efficient reasoning framework that enables two LLMs to collaboratively solve the problem: a long-thought LLM to generate important thoughts, while a short-thought LLM for remaining thoughts, as shown in Table 1. Specifically, we first fine-tune LLMs to follow long-thought reasoning (i.e., `<think>`...`</think>`) and short-thought reasoning (i.e., `<answer>`...`</answer>` or `<answer>`...`</rethink>`) styles under the supervision of synthesized cold start instructions. Moreover, we propose a *Synergizing-oriented Multi-Turn Reinforcement Learning* scheme to further enhance the efficient collaborative reasoning capability of the proposed framework. Different from existing studies Shani et al. (2024) that rely on interactions with the environment to obtain reward signals Zhou et al. (2025); Jin et al. (2025), our long-thought and short-thought LLMs engage in a multi-turn conversation to arrive at a final answer reward. Based on asynchronous policy optimization, the collaboration between long-thought and short-thought LLM could become stable, and their own reasoning capability could be improved, naturally emerging the oha moment Guo et al. (2025); Yang et al. (2025b) of "Overthinking" (e.g., maybe i'm overthinking this. perhaps there's a simpler way) and "Rethinking" (e.g., current approach need to be reconsidered), as illustrated in Table 1.

We validate Long⊗Short on five widely used benchmarks, including MATH 500, AIME 2024, AIME 2025, AMC 2023, and GPQA Diamond. The experimental results demonstrate that Long⊗Short enables current LLM backbones (e.g., Qwen2.5-7B and Llama3.1-8B) to achieve comparable performance compared to their distillation versions (e.g., DeepSeek-R1-Distill-Qwen-7B and DeepSeek-R1-Distill-Llama-8B), while reducing average token length by over 80%.

In summary, our key contributions are threefold: (1) We establish a thought analysis framework, which automatically chunks long CoT into multiple thoughts and quantifies their contributions. (2) We propose Long⊗Short, which adopts cold-start and multi-turn RL training to synergize long-thought and short-thought reasoning. (3) Extensive experiments and theoretical justification demonstrate the effectiveness of our method, offering a foundation for long CoT compression.

Table 1: An illustrative example how a long-thought LLM and a short-thought LLM switch role to answer a given question. In general, long-thought LLM uses `<think>` to start its thought, and use `</think>` to stop thinking process. The short-thought LLM use `<answer>` to continue remaining steps, while use `</answer>` to stop solving process or use `</rethink>` to request the long-thought LLM to generate thoughts again.

| |
|---|
| **Question**: Given that the arithmetic mean of two positive numbers $a$ and $b$ is 4, then the maximum value of the geometric mean of $a$ and $b$ is ? |
| **Ground Truth**: 4 |
| **Long⊗Short**: |
| `<think>` Ok. So I have a question ... `</think>` |
| `<answer>` Let's ... Therefore, current approach need to be reconsidered. `</rethink>` |
| ... |
| `<think>` Wait, ... wait, hold on. maybe i'm overthinking this. perhaps there's a simpler way. `</think>` |
| `<answer>` Therefore, the answer should be 4. `</answer>` |

## 2 PRELIMINARIES

**Definition 1.** *Thought. A thought is the basic logical block (e.g., problem decomposition, solution verification) in a LLM reasoning process. Given a long CoT response $y$, it can be structured as an ordered sequence of thoughts: $y = \{y_1, y_2, \ldots, y_n\}$, where $n$ is the total number of thought.*

Building on this, we further propose the long⊗short thought reasoning paradigm that strategically switches between two LLMs to generate thoughts:

**Problem 1.** *Long⊗Short Thought Reasoning. This work assume that a long CoT reasoning process $y = \{y_1, y_2, \ldots, y_n\}$ can be reformulated into a mixture of long and short thoughts, i.e., $y = \{l_1, s_1, l_2, s_2, \ldots, l_m, s_k\}$, where $l$ represents a important long thought and $s$ denotes a corresponding compressed short thought, while $m$ and $k$ represent their respective counts. The inequality $m + k \leq n$ holds, as multiple thoughts might be compressed into a single short thought.*

This reasoning paradigm enables reasoning process to dynamically alternate between long-thought and short-thought reasoning. As shown in Table 1, the long-thought and short-thought LLMs collaboratively switch roles in order to solve the given question.

## 3 UNDERSTANDING THOUGHTS WITHIN LONG CoT

### 3.1 QUANTITATIVE ANALYSIS BY LONG CoT CHUNKING AND THOUGHT ROLLOUT

Our idea is straightforward: given a question $q$ and the long CoT response $y$ from the reasoning model $\pi_\theta$, we first prompt LLMs to split the long CoT into multiple thought chunks $\{y_1, y_2, \ldots, y_n\}$ shown in Figure 1, each a logical reasoning block (e.g., problem understanding and answer verification). Through rule-based and LLM-based quality evaluation, we find our automated method *achieve 89% accuracy* on long CoT chunking. Prompt template for long CoT chunking is in A.2 and quality analysis details could be found in A.3. Then, we investigate the effectiveness and efficiency contribution of each thought by running Monte Carlo rollouts Snell et al. (2024) on $\pi_{\theta_{base}}$:

**Assumption 1.** *Given a question $q$ and the split thought chunks $y = \{y_1, y_2, \ldots, y_n\}$ from reasoning model $\pi_\theta$, a thought $y_i$ is important if it can help original base model generate a response $y_{base} = \pi_{\theta_{base}}(q, \{y_1, y_2, \ldots, y_i\})$, which achieves higher accuracy with small response length increase.*

With this assumption, we perform a detailed quantitative analysis to investigate how different thoughts influence model effectiveness and efficiency. Specifically, we select DeepSeek-R1-Distill-Qwen-7B Guo et al. (2025) as our long CoT model, which may reflect the accuracy upper bound with all long thought, while using Qwen2.5-7B to indicate the base accuracy and length under the occasion where all thoughts are compressed into short thoughts. We rollout at each thought and report the Pass@1 accuracy and response length. Figure 2 shows on the results on MATH 500 and GPQA Diamond. The following key findings were observed: **(1) The front thoughts bring more accuracy gain:** As shown in Figure 2(a), the accuracy of base model consistently increase when using top-0 to top-10 thoughts to rollout. This trend is consistent across all datasets, indicating that thoughts positioned earlier are more critical for effectiveness. Notably, in the GPQA Diamond dataset, we observe

that the front thoughts (about 8th thoughts) even enable the Qwen2.5-7B to largely outperform DeepSeek-R1-Distill-Qwen-7B on such a totally training-free rollout strategy. These kind of thoughts intuitively should be important. **(2) Long thoughts lead to significant length increase:** As shown in Figure 2(b), the length of base model significantly increase when using thought to rollout. On GPQA Diamond dataset, even only using top-1 thought for base model doubles the response length (from roughly $2.5 \times 10^3$ to $5 \times 10^3$). This reminds us to consider the efficiency of thought, for thought that significantly increase response length to achieve only slight accuracy improvement,

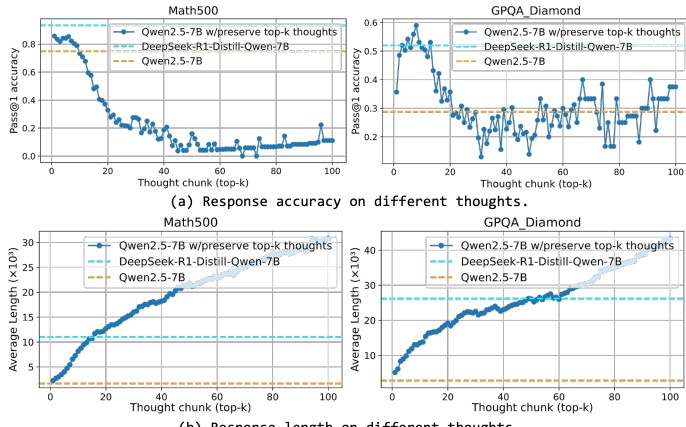

(a) Response accuracy on different thoughts.

(b) Response length on different thoughts.

Figure 2: Quantitative illustration how thoughts affect (a) response accuracy and (b) response length on two reasoning dataset. We investigate how can long thoughts distilled from DeepSeek-R1-Distill-Qwen-7B affect the accuracy and length of base model Qwen2.5-7B. The cyan dashed line can be viewed as the effectiveness upper bound with full long thoughts, whereas the orange dashed line indicates that under full short thoughts.

we may consider to compress these thoughts. We present the cost of the above rollout process in A.4.

Overall, these experimental observations demonstrate that both the accuracy gains and the increased length caused by thoughts are important. It is still a non-trivial problem to justify thought importance based on these scattered analytical evidence.

### 3.2 JOINT MEASUREMENT OF THOUGHT EFFECTIVENESS AND EFFICIENCY

In this section, we propose a unified metric to jointly evaluate the effectiveness and efficiency of each thought. This metric is formulated by considering the accuracy gain from Monte Carlo rollouts alongside factors penalizing excessive length. The metric for thought $y_i$ is defined as:

$$\mathcal{M}(y_i) = \log_2 \left( 1 + \left( \frac{d_y - d_{\{y_1, y_2, \ldots, y_i\}}}{d_y} \right) \cdot \left( \frac{d_y - d_{y_i}}{d_y} \right) \cdot \left( \frac{N_i^{right}}{N_i^{sum}} \right) \right) - \delta(y_i), \quad (1)$$

where (1) $d_y$ is the sum of length of all thoughts; (2) $d_{y_i}$ is the length of the specific thought $y_i$. This term $\frac{d_y - d_{y_i}}{d_y}$ assigns higher scores to shorter thoughts; (3) $d_{\{y_1, \ldots, y_i\}}$ is the cumulative length of thoughts from $y_1$ up to $y_i$. The term $\frac{d_y - d_{\{y_1, \ldots, y_i\}}}{d_y}$ thus assigns higher scores to thoughts with shorter context length; (4) $\frac{N_i^{right}}{N_i^{sum}}$ represents the empirical accuracy obtained by executing Monte Carlo rollouts process $\pi_{\theta_{base}}(q, \{y_1, y_2, \ldots, y_i\})$. $N_i^{sum}$ is the total number of rollouts, and $N_i^{right}$ is the count of correct responses. The higher this term, the greater the effectiveness of thought; (5) $\delta(y_i) \in (0, 1)$ is a conditional decay penalty (set to 0.25 in our experiments). It is activated if the inclusion of thought $y_i$ does not lead to accuracy gain $\frac{N_i^{right}}{N_i^{sum}} \leq \frac{N_{i-1}^{right}}{N_{i-1}^{sum}}$ compared to using thoughts $y_{i-1}$. This term assigns lower score to redundant thoughts, which cannot bring extra accuracy gain.

We present the measurement results in Figure 3, where the total times of rollout increasing from $2^3$ to $2^7$. The results align with our intuition and analysis: high scores are predominantly associated with front thoughts, while

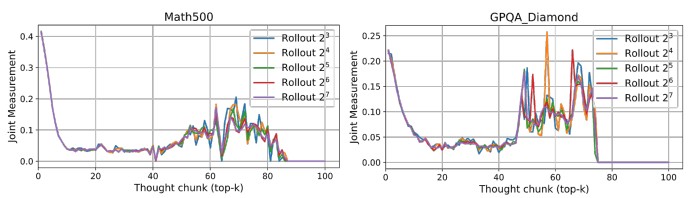

Figure 3: Joint measurement of thought effectiveness and efficiency when Monte Carlo rollouts times increasing from $2^3$ to $2^7$.

only a few rear thoughts occasionally trigger a resurgence in peak values. In addition, as the number of Monte Carlo rollouts increases, the approximate metrics gradually converges to a theoretical curve. For more deep understanding, we analyze the error bound of our proposed measurement.

### 3.3 THEORETICAL JUSTIFICATION

Based on the structure of Equation 1, we assume that the optimal measurement accounts for the true probability of success given the thoughts up to $y_i$, $P_i^{true}$. Let the length-based factors $T_1(i) = \frac{d_y - d_{\{y_1,...,y_i\}}}{d_y}$ and $T_2(i) = \frac{d_y - d_{y_i}}{d_y}$, we can write the optimal measurement as:

$$\mathcal{M}_{opt}(y_i) = \log_2\left(1 + T_1(i) \cdot T_2(i) \cdot P_i^{true}\right) - \delta_{(y_i)}. \tag{2}$$

The error $\mathcal{E}(y_i)$ of our proposed measurement $\mathcal{M}(y_i)$ relative to the theoretical optimum $\mathcal{M}_{opt}(y_i)$ is given by their difference:

$$\mathcal{E}(y_i) = \mathcal{M}(y_i) - \mathcal{M}_{opt}(y_i) = \log_2\left(\frac{1 + T_1(i)T_2(i)\hat{P}_i}{1 + T_1(i)T_2(i)P_i^{true}}\right), \tag{3}$$

where $\hat{P}_i = \frac{N_i^{right}}{N_i^{sum}}$ is the Monte Carlo estimate of $P_i^{true}$. To bound the absolute error $|\mathcal{E}(y_i)|$, we note that it is determined by the term $|\log_2(A) - \log_2(B)|$, where $A = 1 + T_1(i)T_2(i)\hat{P}_i$ and $B = 1 + T_1(i)T_2(i)P_i^{true}$. Using the Mean Value Applying the Mean Value Theorem Siegel (1945), there exists $\xi$ between $A$ and $B$ such that $\log_2(A) - \log_2(B) = (A - B)\frac{1}{\xi \ln 2}$. Thus:

$$|\mathcal{E}(y_i)| \leq |\log_2(A) - \log_2(B)| = \frac{T_1(i)T_2(i)|\hat{P}_i - P_i^{true}|}{|\xi| \ln 2}. \tag{4}$$

Since thoughts have non-zero length ($d_{y_i} > 0$) and the sequence does not cover the full length before thought $n$ ($d_{\{y_1,...,y_i\}} < d_y$ for $i < n$), we have $T_1(i) \in [0, 1)$ and $T_2(i) \in [0, 1)$. Also, $\hat{P}_i, P_i^{true} \in [0, 1]$. Therefore, $A, B \in [1, 2]$, which implies $\xi \in [1, 2]$ and $|\xi| \geq 1$. $|\hat{P}_i - P_i^{true}| \leq \epsilon$ where $\epsilon$ decreases with $N_i^{sum}$. Combining these results, we obtain the following upper bound:

$$|\mathcal{E}(y_i)| \leq \frac{\epsilon}{\ln 2}. \tag{5}$$

This bound reveals that the error is mainly due to the probability error $\epsilon$ of Monte Carlo rollout, and can be reduced by increasing the sample size. The detailed derivation is in the Appendix A.1.

## 4 LONG⊗SHORT

### 4.1 LONG⊗SHORT THOUGHT REASONING COLD START WITH SUPERVISED FINE-TUNING

With the proposed automatic long CoT chunking and joint measurement method, we can obtain a series of pairs containing thoughts and their scores, i.e., $\{y, \mathcal{M}(y)\} = \{(y_1, \mathcal{M}(y_1)), (y_2, \mathcal{M}(y_2)), \ldots, (y_n, \mathcal{M}(y_n))\}$, where $\mathcal{M}(.)$ is the proposed joint measurement of thought effectiveness and efficiency. As shown in Figure 4, the next step is to construct SFT dataset for fine-tuning long-thought and short-thought LLMs.

**Cold Start Data Synthesization.** In this paper, thoughts with high scores are assigned to the long-thought LLM, and those with lower scores are handled by the short-thought LLM. To achieve this, we adopt a heuristic approach via a sequential scan over the trajectory of thoughts $\{(y_1, \mathcal{M}(y_1)), (y_2, \mathcal{M}(y_2)), \ldots, (y_n, \mathcal{M}(y_n))\}$. In this procedure, the first thought, $y_1$, is always preserved as a long thought. For each subsequent thought $y_i$ (for $i > 1$)), we compare its joint measurement $\mathcal{M}(y_i)$ against the maximum score of all previous thoughts: $\mathcal{M}(y_i) > \max\{\mathcal{M}(y_1), \mathcal{M}(y_2), \ldots, \mathcal{M}(y_{i-1})\}$. If this inequality holds, $y_i$ is deemed sufficiently important and is preserved as a long thought. Otherwise, if $\mathcal{M}(y_i) \leq \max\{\mathcal{M}(y_1), \ldots, \mathcal{M}(y_{i-1})\}$, a non-reasoning LLM is prompted to re-complete $y_i$ based on the thought type, and the resulted output naturally serves as a short thought. This procedure ultimately produces a structured sequence comprising alternating thought chunks: $\{l_1, s_1, l_2, s_2, \ldots, l_m, s_k\}$, where $m + k \leq n$ since multiple

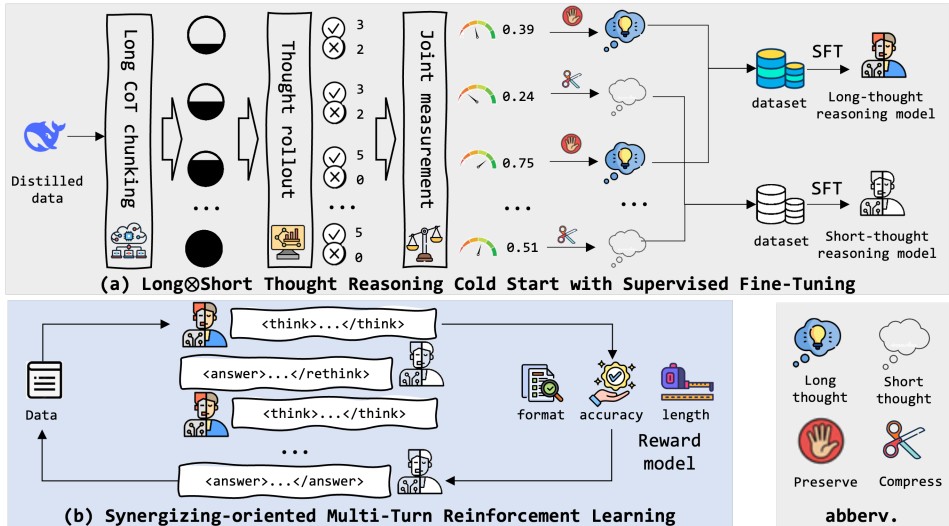

Figure 4: An overview of Long⊗Short framework.

thoughts may merge into a single short thought. We provide prompt template for the proposed thought completion process in A.2 and corresponding quality analysis in A.3. In addition, the cost of cold-start stage and the statistic (e.g., number of chunk) of cold-start dataset could be found in A.4.

**Efficiency-Aware Fine-Tuning.** Subsequently, we concatenate reasoning template and thought context to obtain two SFT datasets $D_{long}$ and $D_{short}$. Specifically, for $D_{long}$, the instruction input $x = [\Gamma_{long} \| q \| h]$ is the concatenation of the long-thought reasoning template $\Gamma_{long}$, question $q$, and historical conversation $h$ (e.g., `<think>`...`</think>``<answer>`...`</rethink>`) between the long-thought and short-thought LLM. While the instruction output corresponds long-thought reasoning process. For $D_{short}$, the instruction input $x = [\Gamma_{short} \| q \| h]$. The difference lies that reasoning template and potential historical conversation $h$ (e.g., `<think>`...`</think>` or `<think>`...`</think>``<answer>`...`</rethink>``<think>`...`</think>`). Detailed reasoning template can be found in A.2. Then, we utilize these datasets to perform full parameter fine-tuning Lv et al. (2023) on a base model:

$$\mathcal{L}_{long/short} = -\mathbb{E}_{(x,o) \sim D_{long}/D_{short}} \left[ \log \mathcal{P}_{\pi_{\theta_{long}}/\pi_{\theta_{short}}} (o \mid x) \right],  \quad (6)$$

where $x$ and $o$ represent the instruction input and output in the reasoning trajectory, respectively. $\mathcal{L}_{long}$ and $\mathcal{L}_{short}$ are the loss of the long-thought LLM and the short-thought LLM. Through this fine-tuning process, we derive two specialized models: $\pi_{\theta_{long}}$, tailored for effectively generate long thought, and $\pi_{\theta_{short}}$, adapted for efficiently producing the short thought.

### 4.2 Synergizing-Oriented Multi-Turn Reinforcement Learning

After the cold-start stage, the long-thought LLM and short-thought LLM can switch roles to collaboratively solve a question through multi-turn conversations. To facilitate more efficient reasoning, we propose a synergizing-oriented multi-turn reinforcement learning framework Shani et al. (2024); Zhou et al. (2025), designed to amplify the effectiveness and efficiency benefits of the long⊗short thought reasoning paradigm.

**Asynchronous Policy Optimization.** Unlike existing multi-turn RL methods Jin et al. (2025); Zhou et al. (2025) that update the model's policy through direct interaction with the environment, our approach requires the long-thought LLM $\pi_{\theta_{long}}$ and the short-thought LLM $\pi_{\theta_{short}}$ to engage in a multi-turn conversation to derive the final reward for policy update. For the $\pi_{\theta_{long}}$, we formulate the optimization goal with a external short-thought LLM $\pi_{\theta_{short}}$ as follows:

$$\max_{\pi_{\theta_{long}}} \mathbb{E}_{x \sim \mathcal{D}, o \sim \pi_{\theta_{long}}(\cdot|x;\pi_{\theta_{short}})} \left[ r(x,o) \right] - \beta \mathbb{D}_{KL} \left[ \pi_{\theta_{long}}(y \mid x; \pi_{\theta_{short}}) \| \pi_{\text{ref}_{long}}(y \mid x; \pi_{\theta_{short}}) \right],$$

$$(7)$$

where $x \sim \mathcal{D}$ represents the input, $o$ is the output generated by $\pi_{\theta_{long}}$ given $x$ and the external $\pi_{\theta_{short}}$, $r(x,o)$ is the reward function, $\pi_{\text{ref}_{long}}$ is the reference LLM for the long-thought LLM, and $\mathbb{D}_{KL}$

denotes the KL-divergence. By explicitly conditioning on an external model during policy learning, it enables more effective collaboration. Symmetrically, when optimizing the short-thought LLM $\pi_{\theta_{short}}$, the $\pi_{\theta_{long}}$ acts as the fixed external LLM:

$$\max_{\pi_{\theta_{short}}} \mathbb{E}_{x \sim \mathcal{D}, o \sim \pi_{\theta_{short}}(\cdot|x;\pi_{\theta_{long}})} \left[ r(x,o) \right] - \beta \mathbb{D}_{\text{KL}} \left[ \pi_{\theta_{short}}(y \mid x; \pi_{\theta_{long}}) \| \pi_{\text{ref}_{short}}(y \mid x; \pi_{\theta_{long}}) \right],$$
(8)

where $\pi_{\text{ref}_{short}}$ is the reference LLM. This alternating optimization ensures that each model update its own policy in synergy with its counterpart, enabling more efficient reasoning.

**Online Sampling Strategy.** To improve policy optimization stability and avoid the need for an additional value function approximation Ramesh et al. (2024), we use Group Relative Policy Optimization (GRPO) Shao et al. (2024) to sample a group of outputs $o = \{o_1, o_2, ..., o_G\}$ for the reference model for each input $x$. The sampling process follows an iterative framework where the system alternates between generating long thoughts (via $\pi_{\theta_{long}}$) and short thoughts (via $\pi_{\theta_{short}}$). This process continues until the short-thought LLM generates a final response, which is enclosed between designated answer tokens, <answer> and </answer>. Then, the reward is computed based on the following hybrid reward model.

**Hybrid Reward Modeling.** We now introduce the reward function $r(x,o)$, which is the primary training signal to guide the optimization process in reinforcement learning. We design a hybrid reward combining correctness, format following, and response length, formulated as:

$$r(x,o) = \eta \cdot \text{EM}\left(a_{\text{pred}}, a_{\text{gold}}\right) + \lambda \cdot \text{FM}(o) + \mu \cdot \text{LM}(o),$$
(9)

where $a_{\text{pred}}$ is the extracted final answer from output $o$ and $a_{\text{gold}}$ is the ground truth answer, the $\text{EM}(.)$ is a rule-based function. $\text{FM}(.)$ represents the format reward function, designed to encourage LLMs to follow the predefined training templates specified in Table 4 and 5 in Appendix A.2. We use the length reward function $\text{LM}(.)$ reported in KIMI K1.5 Team et al. (2025) to guide model use fewer tokens to achieve the final answer. The coefficients $\eta$, $\lambda$, and $\mu$ act as reweighting parameters, enabling control over the contribution of each component in the large-scale reinforcement learning process. We set $\mu = 0$ in the initial stage and gradually increase it. We provide more detailed hyperparameter analysis (e.g., rollout sample size, reward reweighting parameters) in A.7.

## 5 EXPERIMENTS

### 5.1 EXPERIMENTAL SETUP

**Backbone LLMs.** For our experiments, we selected **Llama-3.1-8B**, **Qwen-2.5-7B** and **Qwen3-8B** as our base model, and compare with their long CoT reasoning version, i.e., **DeepSeek-R1-Distill-Llama-8B**, **DeepSeek-R1-Distill-Qwen-7B** Guo et al. (2025), and **Qwen3-8B-thinking**.

**Benchmarks.** We evaluated the performance of our method on five widely used benchmarks: **MATH 500** Hendrycks et al. (2021a), **AIME 2024** and **AIME 2025** Committees (2025), **GPQA Diamond** Rein et al. (2024) and **AMC 2023** Committees (2023).

**Evaluation and Metrics.** We employ the Pass@1 accuracy, average length and Accuracy-Efficiency Score (AES) Luo et al. (2025) as our metrics to assess whether the model achieves a desirable balance between reasoning effectiveness and efficiency. Specifically, $AES = \eta \frac{Length_{base} - Length_{model}}{Length_{base}} + \varsigma \left| \frac{Acc_{model} - Acc_{base}}{Acc_{base}} \right|$, where $Length_{base}$ and $Acc_{base}$ refer to the response token length and accuracy of the distilled long CoT LLMs, respectively. Following original setting in O1-Pruner Luo et al. (2025), we set $\eta = 1$ and $\varsigma = 3$ when $\frac{Acc_{model} - Acc_{base}}{Acc_{base}} \geq 0$, and $\varsigma = -5$ otherwise.

**Implementation Details.** We sample 0.1% of the OpenMathInstruct Toshniwal et al. (2024) dataset—1.8K mathematical problems with ground-truth answer—to distill long CoT responses from DeepSeek-R1, and then prompt the Qwen2.5-72B-Instruct model to perform automatic long CoT chunking. By running Monte Carlo rollouts (5 times per thought) on a small LLM Qwen2.5-7B-Instruct and using Qwen2.5-72B-Instruct to complete short thought, we obtain an SFT dataset (1.4K samples total) for long-thought and short-thought reasoning model. For large-scale multi-turn RL training, we utilize MATH-ligtheval (7.5K training samples and 5K validation samples in total) to conduct iterative self-evolution training. Training details can be found in Appendix A.4.

### 5.2 MAIN RESULTS

Table 2: Comparison of Long⊗Short in asynchronous evolution rounds shows a continual improvement in Pass@1 accuracy and a significant decrease in response length. We denote our model after the cold-start SFT stage as Long-r0⊗Short-r0, where the index indicates the round of policy evolution.

| Round# | MATH 500 Pass@1 ↑ | AMIE 2024 Pass@1 ↑ | AMIE 2025 Pass@1 ↑ | GPQA Diamond Pass@1 ↑ | AMC 2023 Pass@1 ↑ | Avg. Length ↓ | Avg. AES ↑ |
|---|---|---|---|---|---|---|---|
| DeepSeek-R1-Distill-Qwen-7B | **93.40** | **53.33** | **36.67** | **48.98** | **95.00** | 24,566 | 0 |
| Base (Qwen2.5-7B) | 74.80 | 16.67 | 7.25 | 37.88 | 42.50 | 1,623 | -1.33 |
| Long-r0⊗Short-r0 w/SFT | 80.40 | 26.67 | 13.33 | 44.44 | 67.50 | 7,323 | -0.75 |
| Long-r1⊗Short-r0 | 84.60 | 36.67 | 23.33 | 45.45 | 72.50 | 2,169 | -0.08 |
| Long-r2⊗Short-r0 | 87.60 | 43.33 | 26.67 | 46.46 | 85.00 | 2,331 | 0.32 |
| Long-r2⊗Short-r1 | 87.00 | 33.33 | 26.67 | 45.96 | 82.50 | **2,021** | 0.12 |
| Long-r3⊗Short-r1 | 88.60 | 43.33 | 30.00 | 46.96 | 87.50 | 2,457 | 0.43 |
| Long-r3⊗Short-r2 | 87.80 | 43.33 | 26.67 | 46.96 | 87.50 | 2,116 | 0.38 |
| Long-r4⊗Short-r2 | 89.80 | 46.67 | 33.33 | 47.97 | 90.00 | 2,113 | **0.61** |
| DeepSeek-R1-Distill-Llama-8B | **87.20** | **43.33** | 30.00 | **49.49** | **90.00** | 28,496 | 0 |
| Base (Llama3.1-8B) | 62.80 | 10.00 | 6.67 | 35.86 | 22.50 | 3,713 | -1.83 |
| Long-r4⊗Short-r2 | 86.20 | 40.00 | **33.33** | 46.96 | 87.50 | 2,402 | **0.81** |
| Qwen3-8B-thinking | 88.20 | **53.33** | **50.00** | 58.08 | 77.50 | 21,158 | 0 |
| Qwen3-8B-nothinking | 81.60 | 36.67 | 20.00 | 53.03 | 60.00 | 5,580 | -4.20 |
| Long-r4⊗Short-r2 | **88.50** | 43.00 | 46.67 | **59.09** | 80.50 | **3,105** | **0.66** |

As reported in Table 2. We highlight two key observations: **(1) Cold-start stage establishes a strong initial policy.** To enable the long-thought and short-thought LLMs to switch roles effectively, we perform Monte Carlo rollouts to collect SFT data. After fine-tuning, Qwen2.5-7B achieves significant performance gains: 7.4%, 59.98%,

Table 3: Ablation study of SFT and RL on Qwen2.5-7B.

| Round# | MATH 500 Pass@1 ↑ | AMIE 2024 Pass@1 ↑ | GPQA Diamond Pass@1 ↑ | Avg. Length ↓ |
|---|---|---|---|---|
| Base (Qwen2.5-7B) | 74.80 | 16.67 | 37.88 | 1,623 |
| Prompt wo/SFT | 71.60 | 6.67 | 33.83 | 5,362 |
| SFT w/random | 77.20 | 16.67 | 40.91 | 8,234 |
| SFT w/ours | 80.40 | 26.67 | 44.33 | 7,312 |
| RL w/only short | 82.04 | 33.33 | 40.90 | 1,226 |
| RL w/only long | 88.26 | 40.00 | 46.46 | 6,056 |
| RL w/ours | 89.80 | 46.67 | 47.97 | 2,364 |

83.86%, 17.32%, and 58.82% on MATH500, AIME24/25, GPQA Diamond, and AMC23, respectively, with an increase in average response length from 1,623 to 7,323 tokens. Similarly, Llama3.1-8B shows improvements of 16.88%, 133.30%, 149.93%, 12.67%, and 111.11%, with response length increasing from 3,713 to 6,611. The response length increase is expected as the long-thought LLM distills the long CoT reasoning style from DeepSeek-R1. The excessive reasoning length problem will be alleviated by subsequent RL process. **(2) Multi-turn RL continuously improve reasoning capacity.** As reported, after 4 rounds of evolution in the long-thought LLM and 2 rounds in the short-thought LLM, Long⊗Short enables Qwen2.5-7B, Qwen3-8B and Llama3.1-8B to match the performance of DeepSeek-R1-Distill-Qwen-7B and DeepSeek-R1-Distill-Llama-8B, while reducing reasoning token lengths by over 80%. Long-r3⊗Short-r2 also achieves the best AES metric Hou et al. (2025), confirming that iterative RL training maximizes the efficiency. **(3) Long⊗Short Significantly outperform existing efficient reasoning baselines and hybrid reasoning LLMs.** We conduct comprehensive baseline comparison in A.6, the experimental results demonstrate the advantage of Long⊗Short on effectiveness and efficiency across baselines (e.g., CoD and DAST Shen et al. (2025a)), and existing hybrid reasoning LLMs (Qwen3-8B and gpt-oss-20B). We also provide a detailed efficiency analysis and case study of how obtained Long⊗Short works in A.5 and A.8.

## 5.3 Ablation Study and In-Depth Analysis

**Ablation Study on Cold-Start and RL.** We evaluate the effectiveness through the following variants: (1) *SFT w/random*, which replaces the thought metric with random scoring; and (2) *Prompt wo/SFT*, which directly prompts the base model to adopt long-thought and short-thought reasoning styles without fine-tuning. (3) *RL w/only short* refers to RL training only for short-thought LLM. (4) *RL w/only long* refers to RL training only for long-thought LLM. As reported in Table 3, both of direct prompting or replacing thought metric with random value lead to a significant accuracy drop and length increase, highlighting the necessity of the cold-start stage for the long⊗short thought reasoning paradigm. In addition, training with only long-thought or only short-thought LLMs leads to excessive response length or reduced accuracy, validating the proposed collaborative reasoning architecture.

**Model Evolution Analysis on Asynchronous Multi-Turn RL Training.** During iterative RL, the

long-thought and short-thought LLMs are updated asynchronously. Their policy changes influence each other's behavior and jointly affect the overall performance, as shown in Figure 5. In early training, updating the short-thought LLM degrades performance, highlighting the critical role of the long-thought LLM. To address this, we first evolve the long-thought LLM to round 2 while keeping the short-thought LLM fixed, achieving a significant performance gain. Subsequent updates to the short-thought LLM (to

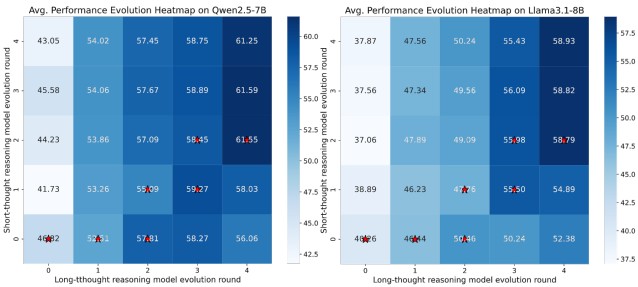

Figure 5: Performance evolution during asynchronous model policy update. We use red stars to indicate our model selection in the 4-round evolution process.

round 1 or beyond) initially cause performance drops, indicating instability in their collaboration. Performance stabilizes and improves as the long-thought LLM evolves further to rounds 3 and 4. Exploring adaptive asynchronous policy optimization is left for future work.

**Aha Moment.** Another phenomenon is the aha moment of 'Overthinking' and 'Rethinking' can be more frequent via RL training process. We provide more details in Appendix A.9.

## 6 RELATED WORK

**Reinforcement Learning for LLM Reasoning.** Recently, reinforcement learning (RL) Ouyang et al. (2022) has been widely adopted to improve LLM reasoning capability. For instance, Proximal Policy Optimization (PPO) Schulman et al. (2017) trains a reward model on human preferences and fine-tunes the LLM policy accordingly. While effective, PPO depends heavily on human-annotated preference data. To simplify training, methods like DPO Rafailov et al. (2023) and SimPO Meng et al. (2024) have been introduced for direct policy optimization without preference annotation requirements. More recently, GRPO Ramesh et al. (2024) further simplifies this pipeline by removing the critic model, instead estimating policy value based on a group of rollout responses. However, these single-turn RL methods cannot interact with environments Casper et al. (2023), limiting multi-step LLM reasoning. To this end, the multi-turn RL Shani et al. (2024) is proposed to improve LLM reasoning by enabling collaboration with external tools Jin et al. (2025) and LLMs. Nevertheless, the application of multi-turn RL to facilitate efficient reasoning remains unexplored.

**Efficient Chain-of-Thought Reasoning in LLMs.** Long chain-of-thought (CoT) reasoning emerged in recent LLMs has demonstrated significant improvement on complex reasoning tasks. To improve reasoning efficiency Jin et al. (2024), recent work focuses on reducing CoT length without sacrificing accuracy Sui et al. (2025), which can be mainly divided into three categories: (1) *Training-free* methods explicitly prompt LLMs to generate limited outputs Kumar et al. (2025) or enforce hard token limits during inference Team (2025). They avoid parameter updates and rely on external constraints to guide brevity Yu et al. (2025b). (2) *SFT-based* methods first apply rejection sampling to obtain correct but shorter reasoning trajectories. Then, approaches like TOPS Yang et al. (2025c) and C3oT Kang et al. (2025) directly use such trajectories for SFT Ma et al. (2025). This strategy encourages LLM to conduct more concise reasoning Shen et al. (2025b). (3) *RL-based* methods employ both offline and online learning strategy. Offline methods like DAST Shen et al. (2025a) construct shorter-longer response pairs Chen et al. (2024) for CoT-length preference learning, while online methods such as Kimi k1.5 Team et al. (2025) and DAPO Yu et al. (2025a) incorporate length rewards to penalize excessive reasoning Arora & Zanette (2025). However, these methods equally compress all thoughts within long CoT, limiting more efficient reasoning.

## 7 CONCLUSION, LIMITATION AND FUTURE WORK

In this work, we propose a theoretically grounded thought analysis framework that incorporates an automatic chunking method and a Monte Carlo thought rollout process to quantify thought importance. Moreover, we introduce Long⊗Short, a collaborative reasoning framework that sequentially execute SFT for reasoning style cold-start and multi-turn RL for model self-evolution, enabling two LLMs to dynamically switch between long-thought and short-thought reasoning styles. Extensive experiments show that Long⊗Short significantly enhances the performance of base LLMs while maintaining competitive efficiency. However, our method incurs substantial computational cost. Future work includes scaling long⊗short thought reasoning with size-varying LLMs.

## ETHICS AND REPRODUCIBILITY STATEMENT

**Ethics statement.** All datasets used in this paper are publicly available and do not contain personally identifiable information. Therefore, this work does not raise specific ethical concerns. **Reproducibility statement.** We release code and data in the supplementary materials to facilitate reproducibility. Researchers can use the provided resources to replicate our experiments and results.

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

# Not All Thoughts are Generated Equal: Efficient LLM Reasoning via Synergizing-Oriented Multi-Turn Reinforcement Learning
### *Supplementary Material*

# Contents

## A APPENDIX

### A.1 THEORETICAL JUSTIFICATION

We first give the mean value theorem lemma Siegel (1945) used for our theoretical justification:

**Lemma 1** (Mean Value Theorem). *Let $f : [a, b] \to \mathbb{R}$ be a continuous function on the closed interval $[a, b]$ and differentiable on the open interval $(a, b)$. Then there exists a point $c \in (a, b)$ such that*

$$f'(c) = \frac{f(b) - f(a)}{b - a}. \tag{10}$$

With the defined lemma, we now analyze the error of our proposed measurement. we first postulate a form for the theoretical optimal measurement $\mathcal{M}_{opt}(y_i)$. Based on the structure of Equation 1, we assume that the optimal measurement accounts for the true probability of success given the thoughts up to $y_i$, $P_i^{true}$. Let the length-based factors $T_1(i) = \frac{d_y - d_{\{y_1,...,y_i\}}}{d_y}$ and $T_2(i) = \frac{d_y - d_{y_i}}{d_y}$, we can write the optimal measurement as:

$$\mathcal{M}_{opt}(y_i) = \log_2 \left( 1 + T_1(i) \cdot T_2(i) \cdot P_i^{true} \right) - \delta_{(y_i)}. \tag{11}$$

Then, The error $\mathcal{E}(y_i)$ of our proposed measurement $\mathcal{M}(y_i)$ relative to the theoretical optimum $\mathcal{M}_{opt}(y_i)$ is given by their difference:

$$\begin{aligned}
\mathcal{E}(y_i) &= \mathcal{M}(y_i) - \mathcal{M}_{opt}(y_i) \\
&= \left[ \log_2 \left( 1 + T_1(i)T_2(i)\hat{P}_i \right) - \delta(y_i) \right] - \left[ \log_2 \left( 1 + T_1(i)T_2(i)P_i^{true} \right) - \delta_{(y_i)} \right] \\
&= \left[ \log_2 \left( 1 + T_1(i)T_2(i)\hat{P}_i \right) - \log_2 \left( 1 + T_1(i)T_2(i)P_i^{true} \right) \right] + \left[ \delta_{(y_i)} - \delta(y_i) \right] \\
&= \log_2 \left( \frac{1 + T_1(i)T_2(i)\hat{P}_i}{1 + T_1(i)T_2(i)P_i^{true}} \right),
\end{aligned} \tag{12}$$

where $\hat{P}_i = \frac{N_i^{right}}{N_i^{sum}}$ is the Monte Carlo estimate of $P_i^{true}$. This error expression shows that the total error due to estimating the true probability $P_i^{true}$ via Monte Carlo rollouts. Then, we derive an upper bound for the absolute error $|\mathcal{E}(y_i)|$. Using the triangle inequality, we have:

$$|\mathcal{E}(y_i)| \leq \left| \log_2 \left( \frac{1 + T_1(i)T_2(i)\hat{P}_i}{1 + T_1(i)T_2(i)P_i^{true}} \right) \right|. \tag{13}$$

Let $A = 1 + T_1(i)T_2(i)\hat{P}_i$ and $B = 1 + T_1(i)T_2(i)P_i^{true}$. The first term is $|\log_2(A) - \log_2(B)|$. Applying the Mean Value Theorem Siegel (1945) shown in Equation 1, there exists $\xi$ between $A$ and $B$ such that $\log_2(A) - \log_2(B) = (A - B)\frac{1}{\xi \ln 2}$. Thus,

$$|\log_2(A) - \log_2(B)| = |A - B|\frac{1}{|\xi| \ln 2}, \tag{14}$$

The difference $|A - B|$ is given by:

$$|A - B| = |(1 + T_1(i)T_2(i)\hat{P}_i) - (1 + T_1(i)T_2(i)P_i^{true})| = T_1(i)T_2(i)|\hat{P}_i - P_i^{true}|. \tag{15}$$

Since $T_1(i) = \frac{d_y - d_{\{y_1,...,y_i\}}}{d_y}$ and $T_2(i) = \frac{d_y - d_{y_i}}{d_y}$ are ratios of lengths within long CoT, and assuming thoughts have non-zero length ($d_{y_i} > 0$) and the sequence does not cover the full length before thought $n$ ($d_{\{y_1,...,y_i\}} < d_y$ for $i < n$), we have $T_1(i) \in [0, 1)$ and $T_2(i) \in [0, 1)$. Also, $\hat{P}_i, P_i^{true} \in [0, 1]$. Therefore, $A, B \in [1, 2]$, which implies $\xi \in [1, 2]$ and $|\xi| \geq 1$. Substituting these into the inequality for the log term:

$$|\log_2(A) - \log_2(B)| \leq \frac{T_1(i)T_2(i)|\hat{P}_i - P_i^{true}|}{1 \cdot \ln 2} \leq \frac{|\hat{P}_i - P_i^{true}|}{\ln 2}, \tag{16}$$

Table 4: Training template for long thought reasoning model.

A conversation between the User and two Assistants. The User asks a question, and two Assistants collaborate to solve it. The first assistant tackles the hard steps, carefully thinks about the reasoning process in the mind, and then provides the reasoning process. The second assistant follows the provided reasoning process to complete the remaining straightforward steps to arrive at the final answer.

Two assistants switch roles to solve the question. The first assistant uses `<think>` to start its thought, and uses `</think>` to stop the thinking process. The second assistant user `<answer>` to complete the remaining steps, and use `</answer>` to stop the solving process. But if the second assistant finds the reasoning insufficient or encounters an error, they use `</rethink>` to request the first assistant to generate thoughts again.

The process is enclosed within `<think>`...`</think>`, `<answer>`...`</rethink>` and `<answer>`...`</answer>` tags, respectively, e.g., `<think>` the first assistant's reasoning process here `</think>` `<answer>` the second assistant's answer here `</rethink>` `<think>` the first assistant's reasoning process here `</think>` `<answer>` the second assistant's answer here `</answer>`.

**You are the first assistant, you return reasoning process starts from `<think>`, and ends with `</think>`.**

where the term $|\hat{P}_i - P_i^{true}|$ represents the absolute error in the Monte Carlo estimation of the success probability. For a sufficiently large number of rollouts $N_i^{sum}$, this error is expected to be small and is bounded. By concentration inequalities (e.g., Hoeffding's), w.h.p., $|\hat{P}_i - P_i^{true}| \leq \epsilon$ where $\epsilon$ decreases with $N_i^{sum}$. Combining the bounds for both terms, the total error is bounded by:

$$|\mathcal{E}(y_i)| \leq \frac{|\hat{P}_i - P_i^{true}|}{\ln 2}. \tag{17}$$

Using the trivial bound $\epsilon$ for the probability error, we obtain an upper bound for the absolute error:

$$|\mathcal{E}(y_i)| \leq \frac{\epsilon}{\ln 2}. \tag{18}$$

This bound reveals that the error is mainly due to the probability nature of Monte Carlo rollout, and can be effectively reduced by increasing the sample size $N_i^{sum}$.

### A.2 PROMPT TEMPLATE

This section introduces the design of the training template and the prompt used to enable automatic Long CoT chunking and short thought completion.

#### A.2.1 TRAINING TEMPLATE

As shown in Table 4 and Table 5, we display the training template for long-thought LLM and short-thought LLM in multi-turn RL training process. In general, these training template guide two LLMs follow tailored style to collaboratively reasoning.

#### A.2.2 AUTOMATIC LONG CHAIN-OF-THOUGHT CHUNKING

Table A.2.2 shows the prompt template used for automatic long CoT chunking. We also provide a chunk example for a mathematical question "Given that $47^{-1} \equiv 51 \pmod{97}$, find $28^{-1} \pmod{97}$, as a residue modulo 97. (Give a number between 0 and 96, inclusive.)". As can be seen the Table A.2.2, we will first split Long CoT into multiple steps by '\n\n', and then we explicitly prompt LLM the chunk long CoT into multiple thoughts. In general, the chunked thought represent the key logical block in the overall long CoT reasoing process.

Table 5: Training template for the short thought reasoning model.

A conversation between the User and two Assistants. The User asks a question, and two Assistants collaborate to solve it. The first assistant tackles the hard steps, carefully thinks about the reasoning process in the mind, and then provides the reasoning process. The second assistant follows the provided reasoning process to complete the remaining straightforward steps to arrive at the final answer.

Two assistants switch roles to solve the question. The first assistant uses `<think>` to start its thought, and uses `</think>` to stop the thinking process. The second assistant user `<answer>` to complete the remaining steps, and use `</answer>` to stop the solving process. But if the second assistant finds the reasoning insufficient or encounters an error, they use `</rethink>` to request the first assistant to generate thoughts again.

The process is enclosed within `<think>`...`</think>`, `<answer>`...`</rethink>` and `<answer>`...`</answer>` tags, respectively, e.g., `<think>` the first assistant's reasoning process here `</think>` `<answer>` the second assistant's answer here `</rethink>` `<think>` the first assistant's reasoning process here `</think>` `<answer>` the second assistant's answer here `</answer>`.

**You are the second assistant, you return completed remaining steps: (1) start from `<answer>`, and end with `</rethink>`, or (2) start from `<answer>`, and end with `</answer>`.**

### A2.2.2 Prompt for Automatic Long CoT Chunking

You are a professional math teacher. When solving math problems, you typically begin by understanding the problem, then break down the solution into several major steps, solving each part in sequence, arriving at the final result, and performing a second verification. Next, I will give you a student's solution process to a math problem. Based on this process, you need to reconstruct their general thinking process during problem-solving — for example, first understanding the problem, then decomposing it, and so on.

Specifically, please first read the problem and their solution. Then summarize the student's thinking process into distinct blocks such as "problem understanding" and so on. Record the start and end step index for each block. Return your answer in JSON format, where the key is "block 1", "block 2", etc., and the value includes the fields start, end, and block type. The start and end should refer to the index of each step, and block type should be a broad category (not too specific), such as "problem understanding".

**Problem:**
{Problem}

**Solution:**
{Long CoT}

Fill in start and end using step indices, and write block type. Return only the following JSON format without any explanations: 'block 1': 'start': '', 'end': '', 'block type': '', ...

**A2.2.2 Example for Automatic Long CoT Chunking**

**Problem:**
Given that $47^{-1} \equiv 51 \pmod{97}$, find $28^{-1} \pmod{97}$, as a residue modulo 97. (Give a number between 0 and 96, inclusive.)

**Solution:**
**Step 0:** So I have this problem here: given that $47^{-1} \equiv 51 \pmod{97}$, I need to find $28^{-1} \pmod{97}$, and it should be a number between 0 and 96, inclusive. Alright, let's see how to approach this.
**Step 1:** First off, $47^{-1} \pmod{97}$ means the multiplicative inverse of 47 modulo 97, which is a number that, when multiplied by 47, gives a result that's congruent to 1 modulo 97. They've already told me that $47^{-1} \equiv 51 \pmod{97}$, so that's a starting point.
**Step 2:** Now, I need to find the inverse of 28 modulo 97. That is, find a number $x$ such that $28x \equiv 1 \pmod{97}$. I wonder if there's a way to relate this to the inverse of 47 that's already given.
**Step 3:** One thing I remember is that in modular arithmetic, if you have inverses of certain numbers, you might be able to combine them in useful ways. Maybe I can express 28 in terms of 47 or something related.
**Step 4:** Let me think about the relationship between 28 and 47 modulo 97. If I can find some expression that relates them, I might be able to use the known inverse of 47 to find the inverse of 28.
**Step 5:** Hmm, perhaps I can find a multiple of 47 that is congruent to 28 modulo 97. That is, find some integer $k$ such that $47k \equiv 28 \pmod{97}$. If I can find such a $k$, then maybe I can manipulate that equation to find the inverse of 28.
**Step 6:** Let's try to solve $47k \equiv 28 \pmod{97}$. Since I know that $47^{-1} \equiv 51 \pmod{97}$, I can multiply both sides of the equation by 51 to solve for $k$:
**Step 7:** $k \equiv 28 \times 51 \pmod{97}$
**Step 8:** Let me calculate $28 \times 51$:
**Step 9:** $28 \times 50 = 1400$
**Step 10:** $28 \times 1 = 28$
**Step 11:** So, $1400 + 28 = 1428$
**Step 12:** Now, compute $1428 \pmod{97}$. To find this, I'll divide 1428 by 97 and find the remainder.
**Step 13:** First, $97 \times 10 = 970$, and $97 \times 4 = 388$, so total $970 + 388 = 1358$.
**Step 14:** Subtracting: $1428 - 1358 = 70$
**Step 15:** So, $1428 \equiv 70 \pmod{97}$, which means $k \equiv 70 \pmod{97}$.
$\dots$
**Step 144:** So, my final answer is that $28^{-1} \equiv 52 \pmod{97}$.
**Step 145:** Final answer: $\boxed{52}$

**Chunked Thought:**
{ "block 1": {"start": 0, "end": 5, "block type": "Question understanding"}, "block 2": {"start": 6, "end": 15, "block type": "Preliminary attempt"}, "block 3": {"start": 16, "end": 25, "block type": "Verify preliminary attempt"}, "block 4": {"start": 26, "end": 49, "block type": "Solve using extended Euclidean algorithm"}, "block 5": {"start": 50, "end": 69, "block type": "Try to solve using known inverse"}, "block 6": {"start": 70, "end": 81, "block type": "Verify answer"}, "block 7": {"start": 82, "end": 142, "block type": "Further attempts and verification"}, "block 8": {"start": 143, "end": 145, "block type": "Final conclusion"} }

### A.2.3 THOUGHT COMPLETION

As shown in Table A.2.3, we present the prompt template used for thought completion, where a sequence of thoughts is completed based on the given thought type. The illustrative example in Table A.2.3 demonstrates that, during an incomplete solution process, we provide the corresponding thought types for the reasoning steps that need to be completed. Thought chunks that already include the reasoning process are regarded as long-thoughts, as previously discussed. We can see that, after thought completion the solution process will be merged by original long thought and completed short thought. These data will be further used for SFT dataset construction.

**A2.2.3 Prompt for Thought Completion**

Given a problem and its corresponding partially completed solution process, please complete the missing parts of the solution based on the specified thought identifiers. Return only the completed results in JSON format. Do not return the steps that have already been provided.
**Problem:**
{Problem}

**Incompleted solution process:**
{Incompleted solution process}

**The parts you need to complete are the content fields corresponding to the following thoughts:**
{Incompleted thought}

Note that the thought numbers are already specified—you must not split or modify them. Please strictly return the completions according to the given numbering, even if the required content for a single thought involves multiple steps.

You must return your answer in the following JSON format, and do not provide any explanation or content for other thoughts: {"Completed Steps": [{"thought": "string", "content": "string"}]}

---

**A2.2.3 Example for Thought Completion**

**Problem:**
Given that the arithmetic mean of two positive numbers $a$ and $b$ is 4, then the maximum value of the geometric mean of $a$ and $b$ is () A: 2 B: 4 C: 8 D: 16

**Incompleted solution process:**
{ "chunk 1": "Ok, so I have a question. This question is about the arithmetic mean and geometric mean. The question says that the arithmetic mean of two positive numbers a and b is 4. We need to find the maximum of their geometric means. The options are a:2, b:4, c:8, d:16.First, I have to recall the definitions of the arithmetic mean and geometric mean. The arithmetic mean is (a + b)/2, and the geometric mean is $\sqrt{ab}$. The question says $(a + b)/2 = 4$, so we can conclude that a + b = 8.", "chunk 2": "Problem decomposition and preliminary solution", "chunk 3": "Wait, wait, I am a little confused, because there are larger numbers in the options, such as 8 and 16. Will there be a larger geometric mean? However, according to the inequality, the arithmetic mean is greater than or equal to the geometric mean, and only when $a = b$, the geometric mean is equal to the arithmetic mean. If $a$ and $b$ are not equal, the geometric mean will be less than the arithmetic mean. Assume that $a \neq b$, for example, $a = 5$, $b = 3$, then $a + b = 8$, the geometric mean is $\sqrt{15} \approx 3.87$, which is indeed less than 4. For another example, if $a = 6$, $b = 2$, the geometric mean is $\sqrt{12} \approx 3.46$, which is still less than 4. It seems that the maximum value of the geometric mean is indeed 4. ", "chunk 4": "Further verification", "chunk 5": "Conclusion" }

**The parts you need to complete are the content fields corresponding to the following thoughts:**
{"Problem decomposition and preliminary solution", "Further verification", "Conclusion"}

**Completed thoughts:**
{ "Completed Steps": [ { "thought": "Problem Analysis and Preliminary Solution", "content": "According to the property that the arithmetic mean is greater than or equal to the geometric mean, the geometric mean is equal to the arithmetic mean only when a = b. The condition given in the question is (a + b)/2 = 4, so a + b = 8. If a = b, then a = b = 4, and the geometric mean $\sqrt{ab} = \sqrt{44} = 4$." }, { "thought": "Further Verification", "content": "Verify whether the geometric mean can be greater than 4 when $a \neq b$. Assuming $a > b$, and given that $a + b = 8$, we can set $a = 8 - b$. Then the geometric mean is $\sqrt{ab} = \sqrt{b(8 - b)}$. Since $b(8 - b)$ is a quadratic function that opens downward, its maximum value occurs when $b = 4$, that is, $a = b = 4$, and the geometric mean is 4. When $b \neq 4$, $b(8 - b)$ will be less than 16, and the geometric mean will also be less than 4. Therefore, the maximum value of the geometric mean is 4." }, { "thought": "Conclusion", "content": "According to the above analysis, the maximum value of the geometric mean is 4, and it can only be achieved when a = b = 4. Therefore, the correct answer is B: 4. " } ] }

---

A.3 QUALITY ANALYSIS OF AUTOMATIC LONG CHAIN-OF-THOUGHT CHUNKING

To assess the reliability of our automatic long Chain-of-Thought (CoT) chunking method, we conduct a quantitative evaluation along two complementary dimensions: (1) rule-based validation of structural and indexing correctness, and (2) LLM-as-Judge assessment of semantic faithfulness to the original reasoning. We report results on the Math500 dataset.

**Rule-based Evaluation.** We validate the generated chunks against a prescribed JSON schema of the form `{"block 1": {"start": "0", "end": "5", "block type": "Question understanding"}, ...}`. We measure: (i) **Format Error**, which flags any deviation from the expected JSON structure; and (ii) **Index Error**, which flags overlapping or inconsistent step indices across blocks (e.g., block 1 ends at step 5 while block 2 starts at step 3).

**LLM-as-Judge Evaluation.** We employ GPT-4o to evaluate whether the chunked thoughts faithfully capture the key logic and reasoning steps of the original long CoT. The judge model is provided with both the original CoT and the chunked representation and prompted to decide whether the latter preserves the original reasoning process: (i) **Faithful**; (ii) **Not Faithful**.

Table 6: Quality analysis of automatic long CoT chunking on Math500.

| Evaluation Method | Metric | Value |
|---|---|---|
| Rule-based | Format Error | 8.4% |
| | Index Error | 1.2% |
| LLM-as-Judge | Faithful (True) | 89.8% |
| | Not Faithful (False) | 0.6% |

Table 7: Statistics of chunked long CoTs

| Dataset | # num of thought | | | # length of thought | | |
|---|---|---|---|---|---|---|
| | min | average | max | min | average | max |
| OpenMath-ThoughtChunk1.8K | 2 | 15.93 | 211 | 8 | 799.3 | 25,188 |
| Math-500-Thought-DeepSeek-R1-Distill-Qwen7B | 2 | 10.67 | 133 | 13 | 929.26 | 17,597 |
| GPQA-Dimaond-Thought-DeepSeek-R1-Distill-Qwen7B | 4 | 14.95 | 132 | 9 | 1,484.60 | 36,842 |
| Math-500-Thought-DeepSeek-R1-Distill-Llama8B | 3 | 12.23 | 220 | 13 | 930.48 | 15,359 |
| GPQA-Dimaond-Thought-DeepSeek-R1-Distill-Llama8B | 4 | 16.12 | 130 | 15 | 1,630.85 | 22,751 |

**Quality Analysis** Table 6 summarizes the quantitative results. We observe that 8.4% of trajectories incur format errors, typically arising in exceedingly long CoTs that stress the output schema. Index errors are rarer (1.2%), indicating that our boundary detection is generally robust. According to the LLM-as-Judge, 89.8% of chunked outputs are judged *Faithful*, whereas only 0.6% are judged *Not Faithful*. The remaining cases are neutral or indeterminate under the judge's criteria. The rule-based errors suggest straightforward avenues for engineering improvements, such as stricter schema validation and adaptive truncation or pagination for extremely long CoTs. The low index error indicates that our automatic boundary detection is reliable in most cases.

## A.4 TRAINING DETAILS OF LONG⊗SHORT

We report the details of the Long⊗Short training process, including dataset construction, resource and costs and the visualization of RL training process of Long⊗Short. We begin by the dataset statistics, and introduce the detailed information about cold-start stage and multi-turn RL training stage.

### A.4.1 DATASET STATISTICS

We report the dataset statistic in cold start stage and multi-turn RL training stage.

**Quantitative Analysis Stage.** In the quantitative analysis stage, we utilize chunked thoughts derived from long chains of thought (CoTs), which are distilled using DeepSeek-R1-Distill-Qwen7B and DeepSeek-R1-Distill-Llama8B. To provide a broader understanding of the structure and scale of the chunked data, we report detailed statistics in Table 7, including the minimum, average, and maximum number of thoughts per CoT, as well as the corresponding statistics for thought lengths (measured by token count). In general, the number of thoughts per CoT ranges from 2 to over 200, with average values around 10–16 depending on the dataset and model. Similarly, the length of individual thoughts varies significantly, from short spans of fewer than 10 tokens to long segments exceeding 30,000 tokens, highlighting the diverse granularity and complexity of reasoning captured in our proposed chunking process.

**Cold Start Stage.** In the cold start stage, we use chunked thoughts obtained from 1.8K long CoTs extracted from the OpenMathInstruct dataset, as detailed in Section 5.1. Table 7 summarizes the key statistics of this chunked data, including the minimum, average, and maximum number of thoughts per CoT and the corresponding lengths of these thoughts. These statistics offer insight into the distribution and scale of reasoning units during the early stage, serving as a foundation for subsequent training or adaptation processes.

**Multi-turn RL training Stage.** For the multi-turn RL traing, we utilize MATH-ligtheval Hendrycks et al. (2021b) (7.5K training samples and 5K validation samples in total) to conduct iterative self-evolution reinforcement learning training.

Table 8: Runtime efficiency on Math500 with a single H800 GPU. We report average response length; average inference latency (s) and average tokens per seconds (Tokens/s).

| Model | Average Response Length | Average Inference Latency (s) | Tokens/s |
|---|---|---|---|
| DeepSeek-R1-Distill-Qwen-7B | 10,774 | 30.28 | 355.81 |
| Base (Qwen2.5-7B) | 1,446 | 4.06 | 356.16 |
| Long⊗Short-Qwen2.5-7B | 1,713 | 4.98 | 343.97 |
| DeepSeek-R1-Distill-Llama-8B | 12,275 | 34.32 | 357.66 |
| Base (Llama3.1-8B) | 3,631 | 10.12 | 358.70 |
| Long⊗Short-Llama-8B | 3,663 | 10.70 | 342.36 |

### A.4.2 RESOURCES AND COSTS IN TRAINING

**Quantitative Analysis Stage.** To enable quantitative analysis of reasoning within long chains-of-thought (CoT), we perform 128 rollouts at each intermediate thought step. For the GPQA-Diamond dataset, this rollout process takes approximately 6 hours using 20 NVIDIA A100 GPUs. For the MATH 500 dataset, the same procedure requires around 10 hours on 20 NVIDIA A100 GPUs.

**Cold Start Stage.** In the SFT cold start stage, we first perform 8 rollouts at each thought step on 1.8K collected long CoTs using Qwen2.5-7B-Instruct, which takes roughly 10 hours on 40 NVIDIA A100 GPUs. Subsequently, we use Qwen2.5-72B-Instruct for thought completion as described in Section 4.1. This step takes approximately 2 hours on 40 NVIDIA A100 GPUs.

**Multi-turn RL Training Stage.** Each round of reinforcement learning (RL) training takes approximately 24 hours on 8 NVIDIA A100 GPUs.

### A.5 INFERENCE EFFICIENCY ANALYSIS OF LONG⊗SHORT

A key practical consideration for collaborative inference is runtime efficiency when switching between two LLMs during generation. Since modern deployments commonly leverage key–value (KV) caching for speed, maintaining two separate caches and recomputing them at switch points may introduce overhead. To quantify this effect, we report detailed throughput metrics, including average response length, average end-to-end inference latency, and decoding throughput (tokens per second), under a standardized setup.

**Setup.** We evaluate all 7B/8B models on a single H800 GPU and collect statistics on the Math500 dataset. For each system, we record: (i) average response length (generated tokens), (ii) average inference latency (seconds) measured end-to-end per instance, and (iii) decoding throughput (tokens/s). We compare the proposed Long⊗Short framework against its corresponding base model, as well as strong distillation-based reasoning models of comparable size.

**Findings.** As reported in Table 8, we observe a small reduction in decoding throughput when employing Long⊗Short, attributable to maintaining two KV caches and recomputing them at switch boundaries. Relative to the base models, throughput decreases by ∼13 tokens/s for Qwen2.5-7B (356.16 → 343.97) and ∼16 tokens/s for Llama-8B (358.70 → 342.36). Despite this overhead, end-to-end latency remains close to the corresponding base models: 4.06 s vs. 4.98 s for Qwen2.5-7B, and 10.12 s vs. 10.70 s for Llama-8B. Crucially, Long⊗Short delivers substantially lower latency than distillation-based reasoning LLMs of similar size (e.g., 30.28 s → 4.98 s for Qwen-7B, and 34.32 s → 10.70 s for Llama-8B) while achieving comparable accuracy.

**Implications and trade-offs.** These results clarify the practical trade-off introduced by collaborative inference: minor decoding slowdowns from dual-cache management versus significant latency reductions relative to distilled long-reasoning models. In typical deployments where end-to-end latency dominates user experience and cost, Long⊗Short offers a favorable balance—retaining base-model responsiveness with only a small throughput penalty. Future work may further mitigate switching overhead via shared-cache layouts, cross-model KV projection, or adaptive batching that aligns switch points across requests.

Table 9: Comparison of Long⊗Short with existing efficient reasoning approaches on Qwen2.5-7B. We selected the average accuracy and length of DeepSeek-R1-Distill-Qwen7B as our baseline for AES metric calculation.

| Methods | | MATH 500 Pass@1 ↑ | AMIE 2024 Pass@1 ↑ | AMIE 2025 Pass@1 ↑ | GPQA Diamond Pass@1 ↑ | AMC 2023 Pass@1 ↑ | Avg. Length ↓ | Avg. AES ↑ |
|---|---|---|---|---|---|---|---|---|
| DeepSeek-R1-Distill-Qwen-7B | | **93.40** | **53.33** | **36.67** | **48.98** | **95.00** | 24,566 | 0 |
| Base (Qwen2.5-7B) | | 74.80 | 16.67 | 7.25 | 37.88 | 42.50 | 1,623 | -1.33 |
| Training-free | CoD | 82.60 | 46.67 | 23.33 | 35.86 | 75.00 | 13,229 | -0.51 |
| | TALE-EP | 88.20 | 50.00 | **36.67** | 41.41 | 85.00 | 16,668 | -0.08 |
| SFT-based | C3oT | 81.00 | 30.00 | 26.67 | 47.47 | 82.50 | 12,576 | -0.42 |
| | DAST | 83.50 | 33.33 | 13.33 | 32.83 | 77.50 | 10,235 | -0.74 |
| | O1-Pruner | 87.55 | 36.67 | 26.67 | 37.87 | 85.00 | 13,467 | -0.37 |
| RL-based | SimplePO-DAST | 82.75 | 33.33 | 26.67 | 43.93 | 77.50 | 17,849 | -0.69 |
| | Kimi k1.5 | 84.00 | 43.33 | 26.67 | 37.37 | 75.00 | 17,583 | - 0.65 |
| Long⊗Short-Qwen2.5-7B | | 89.80 | 46.67 | 33.33 | 47.97 | 90.00 | **2,113** | **0.61** |

Table 10: Comparison of Long⊗Short with existing efficient reasoning approaches on Llama3.1-8B. We selected the average accuracy and length of DeepSeek-R1-Distill-Llama8B as our baseline for AES metric calculation.

| Methods | | MATH 500 Pass@1 ↑ | AMIE 2024 Pass@1 ↑ | AMIE 2025 Pass@1 ↑ | GPQA Diamond Pass@1 ↑ | AMC 2023 Pass@1 ↑ | Avg. Length ↓ | Avg. AES ↑ |
|---|---|---|---|---|---|---|---|---|
| DeepSeek-R1-Distill-Llama-8B | | **87.20** | **43.33** | 30.00 | **49.49** | **90.00** | 28,496 | 0 |
| Base (Llama3.1-8B) | | 62.80 | 10.00 | 6.67 | 35.86 | 22.50 | 3,713 | -1.83 |
| Training-free | CoD | 85.00 | 43.33 | 23.33 | 48.99 | 87.50 | 13,897 | 0.40 |
| | TALE-EP | 79.20 | 40.00 | 30.00 | 41.11 | 82.50 | 18,830 | -0.11 |
| SFT-based | C3oT | 40.20 | 30.00 | 3.33 | 41.41 | 22.50 | 10,014 | -2.06 |
| | DAST | 78.50 | 26.67 | 13.33 | 40.40 | 78.50 | 13,443 | -0.52 |
| | O1-Pruner | 82.50 | 30.00 | 23.33 | 42.93 | 78.50 | 14,486 | -0.22 |
| RL-based | SimplePO-DAST | 82.05 | 26.67 | 23.33 | 37.37 | 82.50 | 16,285 | -0.37 |
| | Kimi k1.5 | 76.50 | 13.33 | 3.33 | 30.30 | 72.50 | 13,684 | -1.21 |
| Long⊗Short-Llama3.1-8B | | 86.20 | 40.00 | **33.33** | 46.96 | 87.50 | **2,402** | **0.81** |

### A.6 Baseline Comparison of Long⊗Short

#### A.6.1 Comparison with Existing Efficient Reasoning Baselines

We also present a comprehensive comparison between our proposed method and existing chain-of-thought long-to-short approaches. Specifically, we categorize the baseline approaches into the following groups:

- **Training-free methods**: These rely on prompt engineering or inference-time techniques without any parameter updates. We select CoD Xu et al. (2025) and TALE-EP Han et al. (2024) as our baselines.

- **SFT-based methods**: These utilize supervised fine-tuning on curated datasets to learn reasoning patterns. We select C3oT Kang et al. (2025), DAST Shen et al. (2025a) and O1-Pruner Luo et al. (2025) as our baselines.

- **RL-based methods**: These leverage reinforcement learning, typically with reward signals derived from task-specific objectives or human feedback, to optimize reasoning performance. We select SimplePO-DAST Shen et al. (2025a), and Kimi k1.5 Team et al. (2025) as our baselines.

As shown in Table 9 and Table 10, our method achieves consistent improvements in both reasoning accuracy and efficiency across multiple benchmarks and model backbones. *(1) On the Qwen2.5-7B backbone*, our method outperforms most baselines in terms of average accuracy while significantly reducing the average output length (2,113 tokens vs. 13K–17K for most baselines). Notably, our approach achieves a superior AES score of **+0.61**, indicating that it not only maintains high reasoning performance but also generates more concise outputs compared to the AES baseline (DeepSeek-R1-Distill-Qwen7B). While some training-free and RL-based methods perform competitively on certain benchmarks, they often incur substantial output length and latency overheads. *(2) On the Llama3.1-8B backbone*, our method again demonstrates strong performance, achieving the highest accuracy on the AMIE 2025 benchmark (33.33%) and the second-best performance on MATH500 (86.20%), while maintaining the lowest average output length (2,402 tokens). Compared to the AES baseline (DeepSeek-R1-Distill-Llama8B), our method yields a notable AES improvement of **+0.81**.

#### A.6.2 Comparison with Existing Reasoning LLMs

In addition, we compare our proposed models, Long⊗Short-Qwen2.5-7B and Long⊗Short-Llama3.1-8B, with both reasoning and non-reasoning models to comprehensively evaluate their effectiveness and efficiency on reasoning tasks. We also compare our method with recently introduced hybrid-thinking-mode LLMs (e.g., Qwen3 Team (2025)). We categorize the baseline LLMs as follows:

- **Non-reasoning models**: These models aim to solve problems quickly without engaging in explicit reasoning. While they offer high speed, the absence of step-by-step reasoning may result in incomplete or less accurate outputs. We select advanced models such as GPT-4-o OpenAI (2024), DeepSeek-V3-671B Guo et al. (2025), Qwen2.5-7B, and Llama3.1-8B as representative examples.

- **Reasoning models**: These models employ extended chain-of-thought capabilities to perform multi-step reasoning across tasks. Although generally more effective, they tend to produce overly lengthy reasoning traces, which can affect efficiency. We include OpenAI-o1 OpenAI (2024), DeepSeek-R1 Guo et al. (2025), QwQ Yang et al. (2024), DeepSeek-R1-Distill-Qwen-7B, and DeepSeek-R1-Distill-Llama-8B as representative models.

- **Hybrid reasoning models**: These models are capable of switching between reasoning and non-reasoning modes, allowing for more flexible inference. We choose Qwen3-8B Team (2025), Qwen3-32B and gpt-oss-20B OpenAI (2025) with low, medium and high reasoning efforts as our comparison models.

As can be seen in Table 11, Long⊗Short demonstrates three key advantages: *(1) Compared with non-reasoning models*, Long⊗Short significantly improves performance across all benchmarks while maintaining similar response lengths. For instance, Long⊗Short-Llama3.1-8B achieves an average of over 20-point gain in most tasks compared to its base Llama3.1-8B with even shorter

Table 11: Comparison with existing advanced LLMs. We bold the results of models with comparable size to highlight the advantages of our efficient inference model over similarly sized LLMs (e.g. Qwen3-8B).

| Methods | MATH 500 Pass@1 ↑ | AMIE 2024 Pass@1 ↑ | AMIE 2025 Pass@1 ↑ | GPQA Diamond Pass@1 ↑ | AMC 2023 Pass@1 ↑ | Avg. Length ↓ | Average Pass@1 ↑ |
|---|---|---|---|---|---|---|---|
| *Non-reasoning model* | | | | | | | |
| Llama3.1-8B | 62.80 | 10.00 | 6.67 | 35.86 | 22.50 | 3,713 | 27.50 |
| Qwen2.5-7B | 74.80 | 16.67 | 7.25 | 37.88 | 42.50 | 1,623 | 35.82 |
| DeepSeek-V3-671B | 90.20 | 39.20 | 30.00 | 69.10 | 90.00 | 3,287 | 63.70 |
| GPT-4-o | 78.60 | 20.00 | 13.33 | 48.48 | 50.00 | 2,840 | 42.08 |
| *Reasoning model* | | | | | | | |
| DeepSeek-R1-Distill-Llama-8B | 87.20 | 43.33 | 30.00 | 49.49 | 90.00 | 28,496 | 60.00 |
| DeepSeek-R1-Distill-Qwen-7B | 93.40 | 53.33 | 36.67 | 48.98 | **95.00** | 24,566 | 65.48 |
| QwQ | 94.40 | 46.67 | 60.00 | 46.67 | 85.00 | 19,504 | 66.55 |
| DeepSeek-R1 | **95.30** | **69.80** | **70.37** | 71.50 | **95.00** | 26,874 | 80.39 |
| OpenAI-o1 | 92.00 | 50.00 | 40.00 | **72.73** | 82.50 | - | 67.45 |
| *Hybrid reasoning model* | | | | | | | |
| Qwen3-8B-thinking | 88.20 | 53.33 | 50.00 | 58.08 | 77.50 | 21,158 | 65.42 |
| Qwen3-32B-thinking | 94.40 | 66.67 | 53.33 | 63.64 | 92.50 | 18,607 | 74.11 |
| gpt-oss-20B-low | 86.60 | 33.33 | 10.00 | 44.94 | 62.00 | 5,158 | 47.37 |
| gpt-oss-20B-medium | 85.20 | 30.00 | 16.67 | 48.48 | 72.50 | 6,026 | 50.57 |
| gpt-oss-20B-high | 92.00 | 36.67 | 46.00 | 56.56 | 88.50 | 14,026 | 63.99 |
| Qwen3-8B-nothinking | 81.60 | 36.67 | 20.00 | 53.03 | 60.00 | 5,580 | 50.26 |
| Qwen3-32B-nothinking | 85.40 | 36.67 | 20.00 | 61.62 | 77.50 | 4,301 | 56.24 |
| Long⊗Short-Llama3.1-8B | 86.20 | 40.00 | 33.33 | 46.96 | 87.50 | 2,402 | 58.80 |
| Long⊗Short-Qwen2.5-7B | 89.80 | 46.67 | 33.33 | 47.97 | 90.00 | 2,113 | 61.55 |

Table 12: Hyperparameters analysis on rollout sample size ($n$).

| $n$ | 1 | 2 | 3 | 4 | 5 | 6 | 7 | 8 |
|---|---|---|---|---|---|---|---|---|
| Accuracy (%) | 75.26 | 77.82 | 77.50 | 79.24 | 80.40 | 80.94 | 81.02 | 81.24 |
| Cost (GPU hours) | 0.93 | 2.18 | 3.14 | 4.07 | 5.12 | 7.48 | 8.52 | 10.24 |

output length (2,402 vs. 3,713 tokens). *(2) Compared with reasoning models*, except for DeepSeek-R1, Long⊗Short achieves comparable or better performance while dramatically reducing token usage—compressing the average output length by more than 80% (e.g., from 24k–28k tokens down to 2k tokens), thus offering a more efficient solution for reasoning-intensive tasks. *(3) Compared with hybrid models*, Long⊗Short consistently matches or surpasses their performance, while generating significantly shorter responses. For example, Long⊗Short-Qwen2.5-7B achieves 90.00 on AMC 2023, outperforming Qwen3-8B-nothinking (60.00) and closely matching Qwen3-32B-thinking (92.50), with less token length than Qwen3-8B-nothinking.

Furthermore, our method achieves the highest average Pass@1 among all non-reasoning models of similar size, surpassing GPT-4-o, Qwen2.5-7B, Llama3.1-8B, Qwen3-8B-nothinking, and even Qwen3-32B-nothinking. The only exception is DeepSeek-V3-671B, which is more than 80× larger than our models. This highlights the strong trade-off between efficiency and performance achieved by Long⊗Short.

## A.7 HYPERPARAMETER SENSITIVITY ANALYSIS OF LONG⊗SHORT

We analyze the sensitivity of Long⊗Short to key hyperparameters used in the SFT cold-start data construction and the subsequent RL stage.

**Rollout sample size ($n$).** As theoretically justified in Eq. (5), the estimation accuracy of the joint measurement depends on the rollout sample size $n$. Increasing $n$ reduces the estimation error $\varepsilon$ but raises computational cost. To quantify the trade-off, we construct SFT cold-start datasets with $n \in 1, \ldots, 8$ and report Math500 performance of Qwen2.5-7B-Cold-Start. As reported in Table 12, accuracy improvements plateau after $n=4$–5, whereas computation grows near-linearly. We therefore select $n=5$ as a cost-effective operating point used in all main experiments. While larger $n$ yields marginal gains, we consider adaptive sampling strategies that allocate larger $n$ only to ambiguous instances an interesting direction for future work.

Table 13: Hyperparameters analysis on RL reward weights $(\eta, \lambda, \mu)$.

| Model / Setting | Accuracy (%) | Length (tokens) | Format Following (%) |
|---|---|---|---|
| Base (Qwen2.5-7B) | 74.80 | 1,446 | – |
| Long-r0 Short-r0 w/ SFT | 80.40 | 5,323 | 78 |
| Long-r1 Short-r0 ($\eta$=1, $\lambda$=1, $\mu$=0) | 84.60 | 1,431 | 95 |
| Long-r1 Short-r0 ($\eta$=0.5, $\lambda$=1, $\mu$=0) | 76.24 | 2,289 | 98 |
| Long-r1 Short-r0 ($\eta$=1, $\lambda$=0.5, $\mu$=0) | 72.04 | 6,857 | 78 |
| Long-r1 Short-r0 ($\eta$=1, $\lambda$=1, $\mu$=0.5) | 81.62 | 1,126 | 93 |

**RL reward weights** $(\eta, \lambda, \mu)$. We ablate the weights governing correctness ($\eta$), structured-format adherence ($\lambda$), and response-length penalty ($\mu$). Our default curriculum keeps the policy focused on correctness while first stabilizing formatting and later compressing length: (i) set $\eta$=1.0 throughout to prioritize accuracy; (ii) set $\lambda$=1.0 for the first 1–2 RL rounds to enforce the structured reasoning format; once format following exceeds $\sim$90(iii) start with $\mu$=0 to permit broad exploration of reasoning strategies, then increase up to 0.5 in later rounds to shorten responses without harming accuracy. As reported in Table 13, we summarize the effects as follows. (1) Correctness weight $\eta$: Lowering $\eta$ from 1.0 to 0.5 substantially degrades accuracy, indicating that correctness must remain the dominant optimization target. (2) Format weight $\lambda$: A high $\lambda$ early stabilizes the structured format and improves overall performance; reducing it too early increases format inconsistency and harms accuracy. After convergence of formatting, a small $\lambda$ prevents reward hacking. (3) Length penalty $\mu$: Introducing $\mu$ too early shortens responses but hurts accuracy. Gradually increasing $\mu$ in later rounds compresses outputs while maintaining performance.

**Takeaways.** Rollout size $n$: choose $n \approx 4$–5 for a strong cost–accuracy trade-off. Rewards: keep $\eta$=1.0; use a high-then-low $\lambda$ to first enforce and then maintain formatting; introduce $\mu$ late to control verbosity without sacrificing accuracy. Future work: adaptive $n$ per-instance and principled, possibly learned, schedules for $(\eta, \lambda, \mu)$ to enhance efficiency and generalization.

### A.8 CASE STUDIES OF LONG⊗SHORT

To gain deeper insights into Long⊗Short, we conduct a case study using Long⊗Short-Qwen2.5-7B, comparing its behavior with DeepSeek-R1. We also provide a breakdown of the number of dialog turns involved in this multi-turn collaborative reasoning process. Finally, we analyze the emergence of the "aha" moment phenomenon during the model evolution process.

#### A.8.1 COMPARISON BETWEEN LONG⊗SHORT AND R1

An illustrative example of how Long⊗Short reasoning works is presented in Table A.8.1. As can be seen, Long⊗Short produces significantly more concise and focused reasoning traces compared to DeepSeek-R1. While R1 generates a long chain of self-reflections—some of which are tangential or redundant—Long⊗Short delivers a more structured and efficient explanation. Moreover, both models ultimately arrive at the correct answer, but their reasoning styles differ: R1 mimics internal dialogue and uncertainty resolution, whereas Long⊗Short is more deterministic and goal-oriented. This illustrates that Long⊗Short can retain reasoning effectiveness while reducing verbosity, suggesting a better alignment with user expectations in scenarios requiring clarity and precision.

Table 14: Detailed statistics of Long⊗Short thought reasoning paradigm.

| Dataset | # num of turn | | | # average length of thought | |
|---|---|---|---|---|---|
| | min | average | max | long-thought | short-thought |
| *Long⊗Short-Qwen2.5-7B* | | | | | |
| Math 500 | 1 | 2.16 | 4 | 799 | 498 |
| AIME 2024 | 1 | 2.47 | 5 | 1,473 | 653 |
| AIME 2025 | 1 | 2.17 | 3 | 1,211 | 620 |
| GPQA Diamond | 1 | 2.13 | 3 | 1,425 | 859 |
| AMC 2023 | 1 | 2.17 | 3 | 991 | 544 |
| *Long⊗Short-Llama3.1-8B* | | | | | |
| Math 500 | 1 | 2.17 | 4 | 1,350 | 906 |
| AIME 2024 | 1 | 2.25 | 4 | 1,340 | 1,224 |
| AIME 2025 | 1 | 2.38 | 4 | 1,453 | 966 |
| GPQA Diamond | 1 | 2.11 | 3 | 1,407 | 859 |
| AMC 2023 | 1 | 2.38 | 4 | 1,032 | 937 |

---

### A.8.1 Case Study 1

**Problem:** 1+1=?

**DeepSeek-R1:**
<think>Okay, so the user is asking what 1 plus 1 equals. Let me think. Well, in basic arithmetic, when you add two numbers together, you're combining their values. So if I have one apple and someone gives me another apple, I have two apples. That makes sense. But wait, could there be any other interpretations here? Sometimes people use 1+1 in different contexts, like binary. In binary, 1 plus 1 would be 10, right? But the question doesn't specify the number system. Since it's a simple question without any context, probably the answer is expecting the standard base 10 result. So 1 + 1 should equal 2. But maybe I should double-check. Let me count on my fingers. One finger up, then add another finger. That's two fingers. Yep, that's right. So unless there's some trick or alternative meaning intended, the answer is 2.</think>
<answer>The result of 1 + 1 is **2**. This is a fundamental arithmetic operation in the base-10 number system. If you encounter this in a different context (e.g., binary, modular arithmetic, or symbolic logic), the answer might vary, but in standard mathematics, the answer is **2**.</answer>

**Long⊗Short:**
<think> Alright, so i've got this problem here: 1+1=? </think> <answer> The problem you've presented is a simple addition problem. When you add 1 and 1 together, the result is 2. So, the answer is 2. </answer>

---

### A.8.2 THOUGHT LENGTH AND DIALOG TURN ANAYLSIS OF LONG⊗SHORT

Table 14 presents a quantitative analysis of the dialog turns and reasoning length under the Long⊗Short paradigm across various math and QA datasets.

Across both Qwen2.5-7B and Llama3.1-8B backbones, we observe that Long⊗Short typically completes reasoning within 2–3 turns, indicating a stable and efficient interaction pattern.

Interestingly, the average length gap between long-thought and short-thought responses suggests the paradigm's ability to modulate verbosity depending on the reasoning role: long-thoughts provide comprehensive problem-solving steps, while short-thoughts summarize or validate the solution. This supports the effectiveness of the Long⊗Short design in encouraging role-specialized reasoning styles.

### A.9 OHA MOMENT OF 'OVERTHINKING' AND 'RETHINKING'

Another phenomenon is the aha moment of 'Overthinking' and 'Rethinking' emerged in RL training process. Although recent works Yang et al. (2025b); Liu et al. (2025) reveal that even base model exhibit aha moment, we find that RL can cause the aha moment to occur frequently, accompanied by a reduction in the response length. As shown in Table 15, we analyze the emergence of oha

Table 15: The comparison of Long⊗Short across asynchronous evolution rounds reveals an increasing percentage (100%) of oha moments generated by the model.

| Round# | MATH 500 | AMIE 2024 | AMIE 2025 | GPQA Diamond | AMC 2023 |
|---|---|---|---|---|---|
| *Long⊗Short-Qwen2.5-7B* | | | | | |
| Long-r0⊗Short-r0 w/SFT | 3.33 | 6.67 | 3.33 | 1.20 | 2.25 |
| Long-r1⊗Short-r0 | 6.67 | 13.33 | 6.67 | 3.33 | 5.23 |
| Long-r2⊗Short-r0 | 15.64 | 36.67 | 16.67 | 9.53 | 15.53 |
| Long-r2⊗Short-r1 | 16.67 | 33.33 | 11.53 | 7.65 | 22.50 |
| Long-r3⊗Short-r1 | 16.50 | 40.33 | 10.54 | 12.50 | 25.25 |
| Long-r3⊗Short-r2 | 14.50 | 42.33 | 16.67 | 14.26 | 24.47 |
| Long-r4⊗Short-r2 | 16.60 | 40.00 | 20.00 | 12.62 | 25.00 |
| *Long⊗Short-Llama3.1-8B* | | | | | |
| Long-r0⊗Short-r0 w/SFT | 3.35 | 0 | 0 | 1.25 | 0 |
| Long-r1⊗Short-r0 | 3.35 | 13.35 | 15.57 | 6.67 | 10.25 |
| Long-r2⊗Short-r0 | 9.76 | 37.54 | 28.50 | 16.67 | 25.50 |
| Long-r2⊗Short-r1 | 16.67 | 42.50 | 30.00 | 17.50 | 33.33 |
| Long-r3⊗Short-r1 | 15.20 | 40.00 | 31.50 | 21.55 | 34.50 |
| Long-r3⊗Short-r2 | 12.50 | 42.50 | 30.00 | 21.00 | 35.55 |
| Long-r4⊗Short-r2 | 14.20 | 43.33 | 33.33 | 22.73 | 37.50 |

moments—self-reflective expressions such as 'Maybe I'm overthinking this' or 'Let me double-check'—across different training stages and datasets shown in Table A.9.

We summarize three key findings: *(1) Initial SFT Models Rarely Exhibit Oha Moments* In the early stage of training (i.e., SFT-only models, e.g., Long-r0⊗Short-r0 w/SFT), the frequency of oha moments is extremely low, often below 5% across all datasets. This indicates that SFT stage alone can only make LLM to learn the basic reasoning style, but the potential reasoning activity is still underactivated. *(2) Reinforcement Learning Significantly Enhances Oha Moment Emergence.* As the models undergo asynchronous multi-turn reinforcement learning, the frequency of oha moments steadily increases. For example, Long⊗Short-Llama3.1-8B reaches over 40% on AMIE 2024 and 2025 datasets by Round 3 and 4. This suggests that RL not only improves task performance but also encourages the model to engage in more reflective reasoning activity in such collaborative reasoning paradigm. *(3) Task Difficulty Correlates with Oha Moment Frequency.* We observe that oha moments are more frequent in datasets with higher complexity, such as AMIE 2024/2025 and GPQA Diamond. On these challenging benchmarks, oha moments appear in up to 40–43% of the model responses in later training rounds. In contrast, simpler datasets such as MATH500 show more modest increases, rarely exceeding 20%. This suggests that oha moments are not uniformly distributed but instead tend to occur when the model faces harder reasoning tasks, indicating that oha moments may serve as a proxy for perceived uncertainty or internal conflict during the long⊗short thought reasoning.

In summary, the progressive rise of oha moments through RL training—particularly on complex datasets—highlights the potential of collaborative and hybrid reasoning paradigm. These findings also suggest a promising direction for developing hybrid reasoning.

## A.9 Case Study 2

**Problem:** Consider a rhombohedral crystal, with the interatomic distance of 10 Angstrom and the angles $\alpha = \beta = \gamma = 30^0$. What is the interplanar distance of the (111) plane of the crystal?

**Long⊗Short:**
`<think>` Alright, I've got this problem about a rhombohedral crystal. So, a rhombohedron is a type of prism where all six faces are rhombuses, and in this case, the interatomic distance is 10 Angstrom, and all the angles $\alpha$, $\beta$, and $\gamma$ are 30 degrees. I need to find the interplanar distance of the (111) plane in this crystal. First, I should recall what interplanar distance means. It's the distance between two parallel planes in a crystal lattice. In this case, I'm looking at the (111) plane, which is a specific plane in the rhombohedral lattice. `</think>` `<answer>` The interplanar distance in a rhombohedral crystal can be calculated using the formula for the interplanar distance in a lattice, which is

$$d_{hkl} = \frac{a}{\sqrt{h^2 + k^2 + l^2 - 2\cos(\alpha)hk - 2\cos(\beta)kl - 2\cos(\gamma)lh}},$$

where $a$ is the lattice constant (10 Angstrom in this case), and $h$, $k$, $l$ are the Miller indices of the plane (111 in this case). The angles $\alpha$, $\beta$, $\gamma$ are all 30 degrees. Plugging in the values, we get ... However, this does not match any of the given options. I will recheck the calculation. `</rethink>` `<think>` Maybe I need to consider the lattice constant differently. Maybe $a$ is not just the interatomic distance, but something else. I need to understand the structure of the rhombohedral lattice. A rhombohedron can be thought of as a cube that's stretched along one axis. In this case, all angles are 30 degrees, which is different from a cube where angles are 90 degrees.... After some calculation, I find that the distance between adjacent atoms along a lattice vector in a rhombohedral lattice with angle 30 degrees is ...This seems incorrect because distance can't be negative. I need to check my signs. After correcting the signs, ... This is close to one of the options, but not exact. I need to verify the calculation. Wait, maybe I'm overthinking on this, the interplanar distance of the (111) plane is 9.54 Angstrom. `</think>` `<answer>` After rechecking the calculation, the correct value for the interplanar distance of the (111) plane in the rhombohedral crystal is approximately 9.54 Angstrom. This matches the candidate answer (1) 9.54 Angstrom. Therefore, the interplanar distance of the (111) plane is 9.54 Angstrom. `</answer>`

### A.10 USAGE OF LARGE LANGUAGE MODELS

In this paper, we primarily use LLMs to plot the statistics figure. We also employ LLMs to identify and correct clear grammatical errors in the authors' drafted paragraphs.

