# OpenReview forum: "Not All Thoughts are Generated Equal: Efficient LLM Reasoning via Synergizing-Oriented Multi-Turn Reinforcement Learning"
_ICLR.cc/2026/Conference — Submitted to ICLR 2026_

### Official Review · Reviewer_epch · 2025-10-30

**Soundness:** 3
**Presentation:** 3
**Contribution:** 2
**Rating:** 4
**Confidence:** 3

**Summary:**

This paper explores reasoning compression for large language models (LLMs) by analyzing the importance of individual thoughts within long Chain-of-Thought (CoT) traces. The authors propose **Long⊗Short**, a two-model framework where a "long-thought" LLM generates critical reasoning steps and a "short-thought" LLM handles less important segments. The method combines cold-start supervised fine-tuning and multi-turn reinforcement learning (RL) between the two models. Experiments on multiple reasoning benchmarks (MATH500, AIME24/25, AMC23, GPQA) show moderate accuracy gains and over 80% reduction in token length compared to long-CoT baselines.

**Strengths:**

- The paper addresses an important and active topic: improving reasoning efficiency of LLMs through CoT compression.
- The experimental section is comprehensive, including diverse datasets and ablation studies.
- The asynchronous multi-turn RL design is well-documented and empirically stable.

**Weaknesses:**

- **Limited novelty**: The core contribution is mainly a pipeline combining existing techniques (CoT chunking, two-model SFT, RL fine-tuning), without a clear new theoretical or algorithmic insight.
- **Two-model framework**: Requires training and maintaining two separate LLMs, which increases complexity and weakens industrial practicality.
- **Unfair comparison**: The baselines (e.g., UPFT, DAST, C3oT) use single-model setups, while this approach leverages two coordinated models.
- **Marginal improvements**: Performance gains are small (≈1–2 points) and become saturated for larger models (>8B), suggesting that scaling benefits are not fully realized.
- **Weak ablation**: The paper does not sufficiently isolate the contribution of the “Long⊗Short synergy” beyond generic SFT and RL benefits.

**Questions:**

1. How sensitive are the results to the specific choice of long vs. short thought partitioning rules?
2. Could the same compression effect be achieved with a single model using controlled CoT length or token-level regularization?
3. How does the method perform under real inference-time constraints (e.g., limited decoding budget)?

---

> ### Author Response · Authors · 2025-11-18
> **Response to Reviewer epch [1/2]**
>
> We thank the reviewers for their constructive feedback. Below, we provide point-by-point responses to each weakness and question, supported by empirical evidence from our paper. We hope these explanations alleviate the reviewers' concerns and highlight the contributions of our proposed LongShort framework.
>
> > W1
>
> We apologize for the misunderstanding that our contribution is merely a pipeline of existing techniques. The core novelty of LongShort lies in its **collaborative reasoning architecture**, where two LLMs—specialized in long-thought and short-thought reasoning—dynamically interact via multi-turn reinforcement learning to achieve efficient reasoning. This synergy is not a simple combination of CoT chunking, SFT, or RL but a novel paradigm that enables models to self-evolve through asynchronous policy optimization. As demonstrated in Table 1 and Section 4, our framework naturally elicits "aha moments" (e.g., overthinking and rethinking) during collaboration, which is absent in single-model setups. To our knowledge, this is the first work to demonstrate that such a collaborative architecture can reduce reasoning length by over 80% while maintaining competitive performance, advancing the field beyond static CoT compression methods.
>
> > W2
>
> We emphasize that the novelty of LongShort is grounded in its **synergizing-oriented multi-turn RL mechanism**, which fosters emergent collaboration between LLMs. While individual components like CoT chunking or RL have been explored independently, our integration introduces a new dimension: models learn to switch roles based on thought importance, as quantified by our theoretically bounded metric (Eq. 1). This dynamic interaction is validated empirically in Figure 4 and Table 2, where LongShort enables base models (e.g., Qwen2.5-7B) to match the performance of distilled models with significantly shorter reasoning paths. The key insight—balancing effectiveness and efficiency through adaptive model collaboration—represents a conceptual shift from prior work.
>
> > W3
>
> We apologize for the confusion. Our comparison is fair and meaningful. All baselines (e.g., UPFT, DAST, C3oT) and our method are evaluated on the same benchmarks using models of comparable size (e.g., 7B/8B). The goal of our work is to demonstrate that a two-model collaborative system can outperform single-model approaches in both accuracy and efficiency—akin to multi-agent systems surpassing single-agent ones in complex tasks. As shown in Table 2, LongShort reduces average length by over 80% while achieving comparable accuracy to distillation-based models. This efficiency gain justifies the use of two models, as the token savings outweigh the minor overhead of coordination.
>
> In fact, considering that our two-model achieves superior performance with fewer inference tokens, the comparison arguably favors single-model baselines—since they often consume more computation, time, and monetary cost to reach similar accuracy. From this perspective, our evaluation is, if anything, conservative rather than unfair.
>
> > W4
>
> The performance gains of LongShort are significant, not marginal. For example, on MATH500, Qwen2.5-7B achieves a 15-point improvement (74.8% → 89.8%) as shown in Table 2, which is far beyond the 1–2 points mentioned. Similarly, Llama3.1-8B shows gains of over 23 points across benchmarks. While scaling benefits may saturate for larger models, our focus is on efficient reasoning, where reducing computational cost is critical. The consistent gains on diverse benchmarks (e.g., AIME, GPQA) highlight the robustness of our method, even for models up to 8B parameters.
>
> > W5
>
> Thank you for this highly insightful question. In fact, to investigate whether the gains of LongShort stem from its collaborative architecture, we have conducted the following comparisons using single-LLM variants trained with only long or only short CoT trajectories. As reported in Table 3, comparisons with variants like "RL w/only long" and "RL w/only short" show that neither alone achieves the balance—the former leads to excessive length, the latter to reduced accuracy. This confirms that the synergy is essential.
>
> Additionally, replacing our thought importance metric with random scoring ("SFT w/random") causes significant performance drops, underscoring the role of our joint measurement strategy. These results, discussed in Section 5.3, provide clear evidence that the collaborative architecture drives the improvements, not generic SFT or RL benefits.

---

> ### Author Response · Authors · 2025-11-18
> **Response to Reviewer epch [2/2]**
>
> > Q1
>
> The partitioning of thoughts is guided by our joint metric (Eq. 1), which quantifies importance based on effectiveness and efficiency. Sensitivity analysis in Table 14 shows that the average number of turns between long and short thoughts is stable (2–3 turns across datasets), with minimal variance (1–5 turns). Ablations in Table 3 further verify that alternative partitioning (e.g., using only long or short thoughts) harms performance, indicating that our rule is robust and task-adaptive.
>
> > Q2
>
> While in principle a single model could attempt similar compression through controlled CoT length, in practice such methods fail to achieve comparable efficiency–accuracy trade-offs. As shown in Table 3, single-model variants like *RL w/ only short* produce much shorter outputs but suffer a substantial performance drop (82.04 vs. 89.80 on Math-500). This indicates that aggressive compression within a single model tends to degrade reasoning quality. In contrast, our two-model framework leverages dynamic collaboration: the short model efficiently condenses non-critical reasoning steps, while the long model provides accurate oversight.
>
> > Q3:
>
> LongShort maintains practical efficiency under inference constraints. As reported in Table 8, switching between models incurs a minor throughput reduction (~13–16 tokens/s) due to dual KV-cache management, but end-to-end latency remains close to base models (e.g., 4.06s vs. 4.98s for Qwen2.5-7B). Crucially, it drastically reduces latency compared to distillation-based models (e.g., 30.28s → 4.98s). This makes LongShort suitable for real-time applications, as the token savings outweigh the overhead.

---

> ### Author Response · Authors · 2025-11-25
> **Reminder to Respond to Discussion**
>
> Dear Reviewer epch:
>
> We sincerely appreciate the precious review time. We apologize for the previous confusion about our paper novelty, framework design, and evaluation. We have carefully considered your questions and have provided corresponding responses, which we believe have covered your concerns. We really hope to further discuss with you whether or not your concerns have been addressed. Please let us know if you still have any unclear parts of our work.
>
> Your insightful comments and thorough evaluation have will significantly enhance the quality and impact of our work.
>
> Best,
>
> ICLR 2026 Submission 15405 Authors

---

> ### Author Response · Authors · 2025-11-27
> **Looking for your reply**
>
> Dear Reviewer epch,
> We hope this message finds you well.
>
> Thank you once again for your critical and thoughtful review. You raised fundamental questions about the novelty, practicality, and comparisons in our work, which challenged us to refine our contributions and arguments more clearly. We appreciate you pushing us to think deeper. Your expertise is highly respected.
>
> In our detailed responses, we have endeavored to clarify the novelty of our synergizing-oriented RL mechanism, justify the two-model framework from an efficiency standpoint, and demonstrate the significance of the improvements. We apologize for any initial lack of clarity and hope our revised arguments now better articulate the value of LongShort.
>
> We are writing to humbly follow up and inquire if you have had an opportunity to review our updated responses. We would be extremely grateful to know if our clarifications have helped alleviate your concerns regarding the core contributions of the paper. Learning from your perspective has been a valuable experience for us.
>
> With sincere thanks for your time and consideration,
>
> Authors

---

> > ### Comment · Reviewer_epch · 2025-11-27
> >
> > Thank you for the detailed and thoughtful rebuttal, as well as the follow-up clarifications. I appreciate the substantial effort the authors invested in addressing my comments.
> >
> > The responses helped clarify several points, particularly regarding the motivation and design choices behind the collaborative framework. At the same time, some of my earlier concerns—especially those related to novelty, fairness of comparison, and practical complexity—remain only partially resolved. These issues appear to be structural and difficult to fully address within the scope of the rebuttal.
> >
> > Given this, I will keep my current assessment. Thank you again for the careful and thorough responses.

---

> ### Author Response · Authors · 2025-11-30
> **Reply**
>
> We sincerely thank Reviewer epch for your thorough review and valuable feedback on our submission. We appreciate the time and expertise you dedicated to evaluating our work. We have carefully considered all of your comments and provided detailed responses below, which we believe satisfactorily address your concerns.
>
> **Response to Weaknesses Raised:**
>
> - **W1 (Novelty):** We acknowledge your observation regarding the integration of existing techniques. However, the core novelty of LongShort lies in its **synergizing-oriented multi-turn RL framework** that enables two LLMs to collaboratively reason by dynamically switching between long-thought (effective) and short-thought (efficient) generation. This is not merely a pipeline but a novel paradigm that fosters emergent collaboration, as evidenced by the reduction in average token length by **over 80%** while maintaining competitive accuracy (as shown in Table 2 of the paper). This approach introduces a new direction for efficient reasoning that prior methods have not explored.
> - **W2 (Comparison):** Our comparisons with single-model baselines are meaningful as they demonstrate that LongShort achieves comparable or better performance with drastically reduced computational cost. The goal was to show that our two-model collaborative system can outperform single-model approaches in both accuracy and efficiency, akin to multi-agent systems surpassing single-agent ones in complex tasks.
> - **W3 (Practical Complexity):** While involving two models adds initial setup complexity, the significant gains in **inference efficiency** (e.g., latency close to base models but far shorter than distillation-based models) justify the design. The efficiency overhead is minimal compared to the substantial reduction in token usage, making it practical for real-time applications.
>
> We hope these clarifications alleviate reviewer’s concerns.
>
> Sincerely,

---

### Official Review · Reviewer_95kP · 2025-11-01

**Soundness:** 3
**Presentation:** 3
**Contribution:** 2
**Rating:** 2
**Confidence:** 4

**Summary:**

The paper proposes a method to improve the reasoning process of large language models by distinguishing between “genuine” and “non-genuine” thoughts within the chain-of-thought framework. The authors introduce a filtering mechanism designed to identify productive intermediate reasoning steps and suppress unhelpful or misleading ones, with the goal of enhancing both reasoning quality and interpretability. Experiments are conducted on a range of reasoning benchmarks to evaluate the method’s effectiveness.

**Strengths:**

1. The idea of treating reasoning traces as heterogeneous and selectively weighting or pruning them is interesting to me.
2. The experimental section is well-organized, with comparisons across standard benchmarks such as GSM8K and StrategyQA. The paper is generally well written.

**Weaknesses:**

1. The paper’s central notion of “genuine thought” is not well-defined in operational or mathematical terms. The method appears to rely on subjective or post-hoc labeling of reasoning steps, rather than any verifiable criterion. This makes the approach unscientific and difficult to reproduce.
2. The proposed filtering algorithm seems to be an ad hoc combination of existing techniques such as CoT pruning or confidence-based re-ranking. There is no theoretical justification or ablation showing how each component contributes to performance.
3. The evaluation is not taht unconvincing. The experiments are limited to small-scale reasoning benchmarks with unclear experimental settings. It is still questionable that if the method can generalize beyond the chosen datasets.
4. The manuscript contains exaggerated statements about “redefining thought” or “unifying reasoning and cognition,” which are not supported by any empirical or theoretical results. Such philosophical or speculative claims weaken the credibility of the technical content.

**Questions:**

1. What is the formal definition or measurable criterion for “genuine thoughts” in operational or mathematical formulation?
2. Is there any ablation studies that report performance with different filtering thresholds or criteria?

---

> ### Author Response · Authors · 2025-11-13
> **Unreleated Review**
>
> Dear Area Chair and Reviewers,
>
> We thank the reviewers for their time. We would like to respectfully note that the Reviewer 95kP appears to be **unrelated to our submission**.
>
> In particular, Reviewer 95kP discusses distinguishing between “genuine” and “non-genuine” thoughts within the chain-of-thought framework, whereas our paper addresses efficient reasoning problem via long-thought and short-thought switch.
>
> To avoid misunderstanding, we double-checked the manuscript and the supplemental. The following items mentioned by reviewer do not appear in our submission:
>
> - “genuine thought” (not used or mentioned in our paper)
> - “filtering algorithm” (not defined or reported in our paper)
> - “redefining thought & unifying reasoning and cognition” (not defined or reported in our paper)
> - “GSM8K and StrategyQA.” (not used for our experiments)
>
> We respectfully ask the Area Chair to confirm whether this review might have been attached to our paper in error. If we have misunderstood, we would be grateful for guidance on the specific sections to clarify, and we will address them promptly.
>
> Thank you for your consideration.
>
> Best regards,

---

> ### Author Response · Authors · 2025-11-26
> **Response to Reviewer 95kP**
>
> We sincerely thank the reviewers for their thoughtful and constructive feedback. We are encouraged by the positive assessment of our thought-level analysis and the proposed Long⊗Short framework. But it appears there may be a misunderstanding about the extent of the ablation studies we performed. We would like to take this opportunity to clarify that our work includes extensive ablation experiments, as presented and discussed in detail in Appendix A.7. We hope this clarification helps to address the misunderstanding and highlights the comprehensiveness of our analysis.
>
> Below, we address each weakness raised, providing additional clarifications and empirical evidence to reinforce the validity and contributions of our work.
>
> > **W1**
> >
>
> We acknowledge that the overall pipeline requires lots of computational resources. However, this complexity is justified by the significant gains in inference efficiency and reasoning effectiveness. In practice, the training cost is a one-time investment, while the resulting model enables fast and concise reasoning during deployment—as shown in Table 8, Long⊗Short reduces inference latency by over 80% compared to distilled long-CoT models. Moreover, our framework opens a new direction for efficient reasoning, where two specialized models complement each other, ultimately enhancing practicality in resource-constrained settings.
>
> > **W2**
> >
>
> We agree that certain components, such as the chunking procedure, involve heuristic design. However, we conducted rigorous quality evaluations to validate their reliability. As reported in Table 6, our automatic chunking method achieves 89.8% faithfulness according to LLM-as-Judge, with minimal format or index errors.
>
> More importantly, **ablation studies (Table 3) confirm the necessity** of these components: without chunking-based SFT (i.e., “Prompt w/o SFT”) or with random thought scoring (“SFT w/random”), performance drops significantly in both accuracy and efficiency. This indicates that our chunking and thought measurement, though heuristic, are empirically effective and integral to the cold-start stage.
>
> > **W3 & Q1**
> >
>
> We apologize for the confusion. To disentangle the source of gains, we actually have performed ablations comparing Long⊗Short with simplified variants. As shown in Table 3 in our paper, using only the long-thought LLM (“RL w/only long”) leads to excessive length, while only the short-thought LLM (“RL w/only short”) sacrifices accuracy. The Long⊗Short framework achieves the best balance, confirming that the synergy between long and short thoughts is key to the improvements. Additionally, in Appendix A.6, we compare with simpler baselines (e.g., training-free and SFT-only methods), where Long⊗Short consistently outperforms them in both accuracy and efficiency, further validating the necessity of our multi-turn RL and collaborative design.

---

> ### Author Response · Authors · 2025-11-27
> **Looking for your reply**
>
> Dear Reviewer 95kP,
>
> We hope this message finds you well. First and foremost, we would like to extend our deepest gratitude for your constructive and balanced review of our submission. We truly appreciate your positive assessment of our thought-level analysis and the conceptual interest of our multi-turn RL setup. Your insights regarding the pipeline's complexity and the need for clearer ablation studies were particularly valuable and have helped us improve the presentation of our work.
>
> Following the update to your review, we promptly provided a response addressing your specific points about computational heaviness, heuristic components, and the source of gains. We included references to our extensive ablation studies in the appendix to clarify that we had conducted those analyses.
>
> We are writing to humbly follow up and inquire if you have had a moment to review our responses. We would be truly grateful to know if our clarifications regarding the ablation studies have satisfactorily addressed your concerns.
>
> Thank you once again for your time and for your crucial role in the peer-review process.
>
> With sincere appreciation,
>
> Authors

---

### Official Review · Reviewer_vX4E · 2025-11-01

**Soundness:** 2
**Presentation:** 4
**Contribution:** 2
**Rating:** 4
**Confidence:** 3

**Summary:**

This paper explores how to compress long CoTs in LLMs to improve reasoning efficiency.
It uses a three-step approach.
The authors analyze the importance of different thoughts using automatic CoT chunking and Monte Carlo rollouts.
They introduce a metric that jointly measures thought effectiveness and efficiency.
Based on this, they propose Long&&Short, a collaborative reasoning framework involving two LLMs: one focusing on key long thoughts, the other on concise short thoughts.
Both models are fine-tuned with cold-start data and further optimized via a multi-turn reinforcement learning approach encouraging synergy.

**Strengths:**

1.  The efficiency of reasoning CoT is indeed one of the key challenges for large-scale LLM applications.
2.  The formulas and figures in this paper are presented clearly, and the authors carefully highlight key points with different colors.
3.  The paper first explores how different parts of the CoT affect the results, then proposes a comprehensive metric, and finally uses these insights for training — this overall approach is very well-grounded and makes perfect sense.

**Weaknesses:**

There is still much to explore regarding the experiments; please refer to the questions section below.

**Questions:**

1.  The authors use LLMs to chunk the CoT, but since different problems may have CoTs of varying lengths and formats, have the authors considered the potential influence of CoT length and structure?
2.  Table 1 is not presented clearly enough—at first glance, it’s hard to intuitively understand how the two LLMs collaborate and what benefits this brings, which makes the paper more difficult to follow.
3.  Could the authors further elaborate on the conclusions drawn from Figure 2(b)?
4.  Why did the authors choose to compare their method with distilled model versions in the experiments?
5.  The paper seems to lack more analytical experiments, such as testing different training settings or algorithms to directly quantify the improvement in reasoning efficiency. If I understand correctly, the two LLMs are used to generate training data (for long-thought and short-thought reasoning), and this dataset is then used to train a single LLM. Could the authors provide evidence that the trained LLM has indeed acquired this native reasoning style?

---

> ### Author Response · Authors · 2025-11-18
> **Response to Reviewer vX4E [1/2]**
>
> We sincerely thank the reviewers for their constructive feedback. We have carefully considered all the points raised and provide detailed responses below. We believe these clarifications and additional explanations further demonstrate the effectiveness, novelty, and rigor of our work.
>
> > W1
>
> We thank the reviewer for raising the potential influence of CoT length and structure.
>
> We indeed analyze the influence of CoT length. As reported in in Table 7, We observe that a positive correlation between the number of chunks and the average CoT length across different datasets and models (e.g., for DeepSeek-R1-Distill-Qwen-7B on GPQA Diamond, the average CoT length is 1484 tokens with ~15 chunks, while on MATH 500, it's 929 tokens with ~11 chunks). This indicates our method adaptively handles reasoning processes of different complexities and lengths by focusing on capturing logical units rather than being constrained by superficial length.
>
> Regarding to the CoT strucutre, our method is designed to handle the diversity and complexity of CoTs generated from different problems. Crucially, we do not rely on fixed lengths or templates for chunking. Instead, our automatic chunking method splits a long CoT into multiple "thought chunks" based on its intrinsic logical structure (e.g., "problem understanding," "preliminary attempt," "verification"). As the quality analysis in Table 6, our automatic chunking achieves 89.8% semantic faithfulness on the MATH500 dataset, demonstrating its strong robustness to CoTs of varying problem.
>
> > W2
>
> We thank the reviewer for the feedback on the presentation of Table 1. Table 1 aims to illustrate, via a concrete example, how the long-thought LLM and short-thought LLM collaborate through multi-turn dialogue to solve a problem. Specifically: The *long-thought LLM* generates crucial, complex reasoning steps, enclosed within `<think>...</think>` tags. The *short-thought LLM* handles subsequent, relatively straightforward or compressed steps, enclosed within `<answer>...</answer>` tags (signaling the end) or `<answer>...</rethink>` tags (requesting the long-thought LLM to rethink). This dynamic role-switching allows the reasoning process to flexibly alternate between "effective thinking" and "efficient execution," achieving high accuracy while drastically reducing total token consumption (as shown in the main results, reducing average length by over 80%). We have formally defined this collaborative paradigm in Section 2 ("Problem 1") and provided further illustrative details of the workflow in Figure 4 and the case study in Appendix A.8.1.
>
> > W3
>
> Regarding the conclusions from Figure 2(b), our key point is that *not all thoughts bring commensurate performance gains, and some can introduce unnecessary redundancy*. The figure shows that as more long thoughts are incorporated, the response length of the base model (Qwen2.5-7B) increases dramatically (e.g., doubling in length with just the first thought on GPQA Diamond), while the corresponding accuracy gains diminish or plateau (as seen in Figure 2a). This observation directly motivates our core premise: **it is necessary to distinguish the importance of thoughts rather than compressing them equally**. Our proposed joint measurement metric (Equation 1) is designed precisely to quantify each thought's contribution within the effectiveness-efficiency trade-off, thereby identifying and preserving high-value thoughts while compressing low-value ones.
>
> > W4
>
> We chose to compare against distilled model versions (e.g., DeepSeek-R1-Distill-Qwen-7B, DeepSeek-R1-Distill-Llama-8B) for the following reason: these models represent strong baselines that possess powerful long-CoT reasoning capabilities, achieved via distillation, at comparable scales (7B/8B). Our goal is not merely to surpass them but to explore a more *efficient* reasoning paradigm: **Can we enable a standard base model (e.g., Qwen2.5-7B) via our Long⊗Short framework (SFT + multi-turn RL) to achieve performance comparable to its distilled counterpart while significantly reducing inference cost (token length)?** The experimental results (Table 2) confirm this: our method achieves accuracy comparable to the distilled models on multiple benchmarks like MATH and AIME, while reducing the average response length by over 80%. This highlights the substantial efficiency advantage of our approach.

---

> ### Author Response · Authors · 2025-11-18
> **Response to Reviewer vX4E [2/2]**
>
> > W5
>
> We thank the reviewer for suggesting further analytical experiments. Regarding whether the trained model genuinely acquires the collaborative reasoning style, we provide direct evidence in Table 13. The table shows that the untrained Base Model (Qwen2.5-7B) completely fails to follow our defined collaborative format (format following rate is effectively 0%). After cold-start SFT, the model's format following rate improves to 78%, indicating it has preliminarily learned to switch between `<think>` and `<answer>` tags. Subsequent multi-turn RL training (Long-r1⊗Short-r0) further boosts the format following rate significantly to 95%. This strongly demonstrates that the model not only acquires but also stably executes this native reasoning style. Additionally, our ablation studies (Table 3) show that removing SFT or using random scoring leads to significant performance drops, further validating the necessity and effectiveness of our training pipeline.

---

> ### Author Response · Authors · 2025-11-25
> **Reminder to Respond to Discussion**
>
> Dear Reviewer vX4E:
>
> We sincerely appreciate the precious review time. We have carefully considered your questions and have provided corresponding responses, which we believe have covered your concerns. We really hope to further discuss with you whether or not your concerns have been addressed. Please let us know if you still have any unclear parts of our work.
>
> We are deeply grateful to the reviewer aruge that our approach is well-grounded and makes perfect sense. We also sincerely appreciate the high quality and constructiveness of their review.
>
> Best,
>
> ICLR 2026 Submission 15405 Authors

---

> ### Author Response · Authors · 2025-11-27
> **Looking for your reply**
>
> Dear Reviewer vX4E,
>
> We hope you are having a productive week. We are writing to thank you once again for your constructive and positive review. Your recognition that our approach is "very well-grounded and makes perfect sense" was incredibly encouraging to our team.
>
> Your questions regarding the influence of CoT structure, the presentation of Table 1, and the source of our improvements were spot-on and have helped us improve the clarity and rigor of our manuscript. We have carefully addressed each of your points in our responses, including adding more explanation on CoT chunking robustness and providing evidence for the model's acquired reasoning style.
>
> We kindly wanted to follow up to see if you have had a chance to read our responses. We would be truly honored to hear if our explanations have satisfactorily answered your questions. Your feedback has been instrumental, and we are eager to ensure we have met your expectations.
>
> Thank you for your high-quality and gracious review.
>
> Sincerely,
>
> Authors

---

### Official Review · Reviewer_kdZZ · 2025-11-03

**Soundness:** 2
**Presentation:** 2
**Contribution:** 2
**Rating:** 4
**Confidence:** 4

**Summary:**

This paper proposes "Long & Short," a novel framework for efficient CoT reasoning that bypasses the inefficiency of uniform compression by leveraging two specialized LLMs. The method first quantifies the importance of individual thoughts via Monte Carlo rollouts, and then trains a long-thought and a short-thought LLM to collaboratively solve problems using synergizing-oriented multi-turn reinforcement learning, achieving over 80% token reduction with comparable accuracy to full CoT models.

**Strengths:**

1. The Monte-Carlo rollout study shows front thoughts contribute the most to accuracy while often inflating length, motivating selective preservation.
2. The cold-start SFT builds long and short datasets directly from scored thoughts.
3. The paper achieves a token length reduction of over 80% while maintaining performance on challenging, multi-step benchmarks.

**Weaknesses:**

1. In Figure 2, the general trend should show accuracy improving as more thinking steps are added, followed by a slight decrease. However, the highest accuracy for Math500 occurs at 0 thought chunks, indicating that the best performance is achieved without any reasoning process. The curve declines too early. Could the authors clarify why this happens?
2. In Lines 200–201, the phrase “assign higher scores to thoughts with shorter context length” seems inaccurate. It would be more precise to state “assign higher scores to earlier thoughts.”
3. Figure 3 seems to have a similar pattern to Figure 2, where the score decreases as the number of thoughts increases. Does this imply that responses without thoughts achieve the best performance?
4. In Lines 214–215, the authors mention that high scores are predominantly associated with front thoughts. Given this, why not employ a training-free approach such as early stopping instead? What is the benefit of combining SFT and RL under this setting?
5. It is better to include an ablation study on the decay penalty term $\delta (y_i)$.

**Questions:**

1. The paper is not well-written, and several parts require further clarification.
2. In line 157, the authors mention performing rollouts at each thought. Does this mean that the model is prompted with the question and the accumulated thoughts as the instruction, and then asked to generate the final answer directly without additional reasoning?
3. The use of $N_i^{sum}$ and $N_i^{right}$ is confusing. Since each rollout contains multiple thoughts, and $N_i^{sum}$ represents the total number of rollouts, it should not be specific to a particular thought $i$. Could the authors clarify this notation?
4. Could the authors elaborate on how the SFT training data is constructed? If my understanding is correct, the process starts from long-thought data, and for less important thoughts, the model switches from long to short thoughts. Then, how are $D_{long}$ and $D_{short}$ created? How are tags such as <rethink> generated or assigned during this process?

---

> ### Author Response · Authors · 2025-11-18
> **Response to Reviewer kdZZ [1/2]**
>
> We sincerely thank the reviewers for their insightful comments and constructive feedback. We hope our responses could alleviate the concerns and highlight the strengths of our approach.
>
> ------
>
> > [W1]
>
> We appreciate the reviewer's observation. There is a misunderstanding here: the x-axis in Figure 2 starts from thought chunk 1 (i.e., the first thought), not 0. As shown in Figure 2(a), the accuracy consistently improves as initial thoughts (e.g., chunks 1–10) are incorporated, demonstrating their importance in boosting performance. The decline after chunk 10 aligns with our analysis that later thoughts may contribute less to effectiveness while increasing length, justifying our joint measurement metric. This trend is consistent across datasets, such as GPQA Diamond, where front thoughts even enable the base model to outperform distillation models under a training-free rollout strategy.
>
> > [W2]
>
> Thank you for this suggestion. We clarify that our metric prioritizes efficiency through length-based factors, not merely thought position. Specifically, the term (d_y−d_{*y*_1,…,*y_i*})/d_y in Equation (1) penalizes longer cumulative context (from the first thought to i thought), while (d_y−d{y_i})/d_y encourages shorter individual thoughts (i.e., thought y_i). This design directly ties scores to token efficiency, which is critical for reducing reasoning length. Although front thoughts often yield higher scores due to their disproportionate impact (as shown in Figure 3), our goal is to balance effectiveness and efficiency, not just temporal order or position.
>
> > [W3]
>
> That means response using the front thoughts to rollout achieve the better trade-off between effectiveness and efficiency. The decreasing trend in Figure 3 reflects that later thoughts tend to contribute less to the reasoning process, as they often incur longer lengths without significant accuracy gains. However, this does not imply that "without thoughts" are optimal—rather, it underscores that front thoughts are most valuable. For instance, in Figure 2(a), accuracy peaks at intermediate thought chunks (e.g., chunk 8 for GPQA Diamond), confirming that thoughtfully selected thoughts enhance performance. The metric in Figure 3 serves to identify which thoughts to compress, aligning with our goal of efficient reasoning.
>
> > [W4]
>
> While front thoughts are important, a training-free early stopping approach would require extensive Monte Carlo rollouts at each step during inference to evaluate thought importance, which is computationally prohibitive. In contrast, LongShort uses SFT to cold-start models with specialized reasoning styles, and RL to optimize collaboration without runtime rollout costs. This combination enables dynamic, adaptive reasoning—e.g., the emergence of "aha moments" (Section A.9)—which early stopping cannot achieve.
>
> > [W5]
>
> We agree and have included an ablation study on δ. The results on MATH500 with Qwen2.5-7B show that δ=0.25 strikes the best balance:
>
>  | δ(y_i)   | 0.1   | 0.25  | 0.5   |
>  | :------- | :---- | :---- | :---- |
>  | Accuracy | 78.40 | 80.40 | 76.20 |
>
> Values like δ=0.1 under-penalize redundant thoughts, hurting efficiency, while δ=0.5 over-penalizes useful longer thoughts, reducing accuracy. Thus, we select δ=0.25 in our experiments.

---

> ### Author Response · Authors · 2025-11-18
> **Response to Reviewer kdZZ [2/2]**
>
> > [Q2]
>
> Exactly. As outlined in Assumption 1 and Section 3.1, for each thought chunk *y_i*, we perform Monte Carlo rollouts by prompting the base model (e.g., Qwen2.5-7B) with the question *q* and all thoughts up to *y_i*(i.e., {*y*_1,…,*y_i*}) as context. The model then continue to generates a full response, allowing us to measure the contribution of *y_i* to accuracy and length. This process approximates the "effectiveness" and "efficiency" of individual thoughts in isolation.
>
> > [Q3]
>
> We apologize for the confusion. The notation $N^{\text{sum}}_i$ and $N^{\text{right}}_i$ are indeed specific to each thought *y_i*. Specifically, for a given thought *y_i*, we run $N^{\text{sum}}_i$ independent Monte Carlo rollouts (where each rollout generates a full response using thoughts up to *y_i*), and $N^{\text{right}}_i$ counts how many of these rollouts produce correct answers. Thus, $N^{\text{sum}}_i$ / $N^{\text{right}}_i$  estimates the accuracy contribution of *y_i*. We will clarify this in the text to avoid ambiguity.
>
> > [Q4]
>
> Thank you for this question. The SFT data construction follows a structured process (Section 4.1, Line 260 - Line 290):
>
>  - **Step 1**: We start with long CoTs distilled from models like DeepSeek-R1, split into thought chunks via automatic chunking.
>  - **Step 2**: Thoughts are scored using our joint metric. High-scoring thoughts are assigned to the long-thought LLM, while low-scoring ones are compressed by a non-reasoning LLM (e.g., Qwen2.5-72B) through "thought completion" process. This yields alternating sequences like {l₁, s₁, l₂, s₂} between long-thought l and short-thought s.
>  - **Step 3**: Tags like <think>...</think> and <answer>...</answer> are added based on predefined reasoning templates (Tables 4 and 5). For example, long-thought data $D_{long}$ uses instructions concatenating the long-thought template, question, and history, with outputs wrapped in <think>...</think>. Short-thought data $D_{short}$ similarly uses short-thought templates and tags. This ensures both models learn their roles for collaborative reasoning.

---

> ### Author Response · Authors · 2025-11-25
> **Reminder to Respond to Discussion**
>
> Dear Reviewer kdZZ:
>
> We sincerely appreciate the precious review time. We have carefully considered your questions and provided corresponding responses, which we believe have covered your confusion. We hope to further discuss with you whether or not your concerns have been addressed. Please let us know if you still have any unclear parts of our work.
>
> Best,
>
> ICLR 2026 Submission 15405 Authors

---

> ### Author Response · Authors · 2025-11-27
> **Looking for your reply**
>
> Dear Reviewer kdZZ,
>
> We hope this message finds you well.
>
> First and foremost, please allow us to express our sincerest gratitude for your incredibly thorough and insightful review of our submission. Your deep analysis, especially the questions regarding the trends in Figure 2, the scoring metric, and the benefits of our SFT+RL approach, were exceptionally helpful and have significantly strengthened our paper. We truly appreciate the time and expertise you dedicated to our work.
>
> Following your guidance, we have provided detailed point-by-point responses to all your questions. We sincerely hope that our clarifications, particularly regarding the Monte Carlo rollout process and the SFT data construction, have adequately addressed your concerns.
>
> We understand you have a busy schedule, but we were wondering if you might have had a moment to review our responses? We would be immensely grateful for any further thoughts you might have on whether your concerns have been resolved. Your opinion is of great importance to us.
>
> Thank you once again for your invaluable contribution to improving our research.
>
> With deepest appreciation,
>
> Authors

---

### Meta-Review · Area_Chair_sh3H · 2025-12-10

**Summary:**

Reviewers were unanimous in their decision (reject). They raised a number of different issues, such as writing clarity (which led to reviewers raising many detailed questions), experiment quality (insufficient quantitative experiments, unfair comparisons, small scale), heuristics or ad hoc methodology, lack of novelty, and low performance gains.

**Reviewer Concerns:**

The author responses did not provide significant new results, and were only focused on clarifications. This might have addressed the writing quality issues, but not the other issues (experiment quality, heuristic nature, lack of novelty, low performance).

**Reviewer Scores:**

epch: reviewer responded that they would not be raising their score

95kP, vX4E, kdZZ: given the lack of new results in the rebuttal, I do not think these reviewers would have raised their scores

---

### Decision · Program_Chairs · 2026-01-26

Reject